# LC3-associated phagocytosis promotes glial degradation of axon debris after injury in *Drosophila* models

Áron Szabó [1] ✉, Virág Vincze[1,5], Aishwarya Sanjay Chhatre [1,2,5], András Jipa [1], Sarolta Bognár [1], Katalin Eszter Varga[1], Poulami Banik [1], Adél Harmatos-Ürmösi [1], Lukas J. Neukomm [3] & Gábor Juhász [1,4] ✉

Glial engulfment of neuron-derived debris after trauma, during development, and in neurodegenerative diseases supports nervous system functions. However, mechanisms governing the efficiency of debris degradation in glia have remained largely unexplored. Here we show that LC3-associated phagocytosis (LAP), an engulfment pathway assisted by certain autophagy factors, promotes glial phagosome maturation in the *Drosophila* wing nerve. A LAP-specific subset of autophagy-related genes is required in glia for axon debris clearance, encoding members of the Atg8a (LC3) conjugation system and the Vps34 lipid kinase complex including UVRAG and Rubicon. Phagosomal Rubicon and Atg16 WD40 domain-dependent conjugation of Atg8a mediate proper breakdown of internalized axon fragments, and Rubicon overexpression in glia accelerates debris elimination. Finally, LAP promotes survival following traumatic brain injury. Our results reveal a role of glial LAP in the clearance of neuronal debris in vivo, with potential implications for the recovery of the injured nervous system.

Neural injuries, stroke and neurodegenerative diseases generate a significant volume of dead cell-derived material that is cleared by phagocytic glia or macrophages. In mammals, debris generated after harmful events and synapses or neurites formed in excess during development, are eliminated by microglia and astrocytes[1–3]. Major trauma, stroke and neurodegeneration can lead to reactive gliosis, which, if not contained in time, can be maladaptive and result in inhibition of axon and synapse regeneration and in neuroinflammation[4]. Initial microglial activation and ensuing debris phagocytosis help to resolve the damage in the affected tissue at the lesion site, thereby reducing the risk of inflammation and secondary neurodegeneration. Therefore, establishing the correct balance between glial activation states and the timing of debris clearance seems crucial for nervous system recovery.

Studies on *Drosophila* have elucidated several aspects of the mechanism of axon debris phagocytosis by glia[3,5–7]. *Drosophila* glial types are functionally similar to glia in mammals[5]. Ensheathing glia are the phagocytic cell type in the adult fly brain[8] while in the adult peripheral nervous system (PNS), wrapping and subperineurial glia were found to engulf most of the axon debris generated after injury[9].

Injury of olfactory sensory neuron axons coupled to genetic screens in glia allowed identification of several components of the debris clearance machinery in flies[6,7]. Adult glia engulf dead cell-derived material through the action of Draper (Drpr), a transmembrane receptor with multiple EGF repeats[6,10] that recognizes phosphatidylserine[11–13]. Drpr initiates Rac1 small GTPase activation through guanine nucleotide exchange factor (GEF) complexes Drk/Dos/Sos and Crk/Mbc/dCed-12[14] that promotes glial extensions toward

[1]Biological Research Center, Institute of Genetics, Eötvös Loránd Research Network (ELKH), Szeged H-6726, Hungary. [2]Doctoral School of Biology, University of Szeged, Szeged, Hungary. [3]Department of Fundamental Neurosciences, University of Lausanne, CH-1005 Lausanne, Switzerland. [4]Department of Anatomy, Cell and Developmental Biology, Eötvös Loránd University, Budapest H-1117, Hungary. [5]These authors contributed equally: Virág Vincze, Aishwarya Sanjay Chhatre. ✉e-mail: aszabo@brc.hu; juhasz.gabor@brc.hu

debris and thereafter debris internalization[15]. Compared to the above olfactory sensory neuron injury data, only Draper's essential role for debris engulfment by wrapping (and to a lesser extent by sub-perineurial) glia has been established in the PNS[9].

While the steps of early activation and the extension of glial protrusions toward debris are relatively well known, whether and how phagocytosis and breakdown of axon debris by glia are regulated remains largely unexplored. Injury of olfactory sensory neuron axons leads to a transient increase in glial lysosome acidification in the region of degenerating axons[15]. Glial phagosomes that form after injury contain Drpr and fuse with acidic lysosomes. The Rac1 GEF complex Crk/Mbc/dCed-12 is not required for glial activation, it mediates internalization of axon debris and Crk or dCed-12 silencing in glia prevents lysosome acidification[15]. The regulatory mechanisms of phagosome maturation and fusion with lysosomes during debris clearance are mostly unknown.

Macroautophagy (hereafter autophagy) is a membrane-contained intracellular degradation pathway that selectively eliminates unwanted cytoplasmic material such as aged or defective organelles and protein aggregates. Autophagy also supplies the cells with recycled nutrients upon starvation by degrading portions of the cytoplasm[16]. Canonical autophagy is initiated by the Atg1 kinase complex that conveys AMPK and Tor signaling-mediated nutritional information to other components, including a Class III phosphatidylinositol-3-kinase (PI3K) lipid kinase complex[16,17]. This prepares membranes with phosphatidylinositol-3-phosphate (PI3P) for expansion of phagophores that finally close to form autophagosomes containing cytoplasmic cargoes. Two ubiquitin-like protein conjugation systems promote phagophore expansion and autophagosome closure. This involves conjugation of Atg8a to phosphatidylethanolamine (PE) on phagophore/autophagosome membranes. Thereafter, Atg8a also contributes to organizing autophagosome fusion with lysosomes[16,17].

The Vps34 complex has different subunit compositions depending on the trafficking process in question. In mammals, Atg14L incorporates into Vps34 complex I to promote autophagy in concert with core subunits VPS34, VPS15 and Beclin 1[18]. UVRAG may also support autophagy, while it is important for endosome maturation and LAP by forming Vps34 complex II through replacement of Atg14L. The autophagic Vps34 complex in Drosophila similarly contains Vps34, Vps15 and Atg6/Beclin 1 as core subunits and Atg14 as a specific subunit[19]. Drosophila UVRAG is dispensable for autophagy in fat cells[20,21] and is required for endosome maturation and crinophagy[21–23].

LC3-associated phagocytosis (LAP) is a non-canonical autophagy-related pathway that contributes to phagocytic clearance in certain conditions. During LAP, Rubicon binding to the UVRAG-containing Vps34 complex redirects it to phagosomes that contain cargo such as dead cells, entotic cells, cell remnants, pathogens, and photoreceptor outer segments that have been recognized by cell surface receptors[24–27]. LC3 lipid conjugation onto the single-membrane phagosome promotes its fusion with the lysosome where the phagosome content is degraded. The importance of LAP in neural tissues is only beginning to be elucidated[28–30] and it is unclear whether LAP contributes to the restoration of normal physiology after nervous system injury.

In this work, we aimed to understand the role of autophagy and related pathways during axon debris clearance after injury. We hypothesized that autophagy genes may play a role in three different and non-exclusive scenarios. First, autophagy induction in degenerating axon fragments could accelerate or limit debris production[31]. Second, autophagic vesicles may fuse with phagosomes to promote the clearance of the engulfed material in glia[32,33]. Lastly, an autophagy-related process such as LAP may be necessary for phagosome maturation and degradation[34]. Here we show that it is LAP that promotes glial degradation of injured axon debris. Atg8a and Rubicon co-localize with phagocytosed axon fragments in glia, and glial loss of a subset of autophagy genes and of Rubicon impairs debris elimination and phagosome-lysosome fusion. Accordingly, this special form of phagocytic degradation underlies efficient removal of axon fragments after peripheral or central nervous system injury in Drosophila.

## Results

### A subset of Atg proteins functions in axon debris clearance

Numerous reports have demonstrated that autophagy can facilitate the clearance of apoptotic cell corpses[32,33]. To investigate whether autophagy participates in the breakdown of axon debris after injury, we evaluated debris elimination in the Drosophila L1 vein wing nerve, an established model for Wallerian degeneration in the PNS following unilateral wing transection[9,35–37] (Figs. 1a and S1a). In this model, about 40 wing nerve axons are usually labeled by OK371-Gal4, UAS-mCD8::GFP to monitor axon fragmentation and subsequent debris removal[9]. Importantly, a small subset of axons always remains uninjured in this wing severing model because their sensory neuron cell bodies (located at the anterior wing margin) are proximal to the cut site (Figs. 1a and S1a). The number of uninjured axons seen in confocal images may vary depending on the exact location of the cut site and whether they are present in the optical plane obtained by microscopy. There was no statistically significant difference in the number of uninjured axons between autophagy mutants and controls after wings were cut in half (Fig. S1c).

Atg8a is the main Drosophila ortholog of mammalian LC3 family proteins[38] that becomes membrane-bound via its covalent conjugation to phosphatidylethanolamine (PE), thanks to the action of an E3-like ligase complex composed of Atg5, Atg12 and Atg16. We first studied axon fragmentation and persistence in the viable $Atg8a^{\Delta4}$ deletion mutant[39] and a trans-heterozygous Atg16 mutant combination ($Atg16^{\Delta67/\Delta129}$). We saw no alterations at 2 days post-injury (dpi) (Fig. 1b, c), while accumulated axon debris persisted in $Atg8a^{\Delta4}$ and $Atg16^{\Delta67/\Delta129}$ wings at 5 and 10 dpi, respectively, unlike in controls where debris was gradually cleared (Fig. 1b, c). Axon debris clearance could be fully restored in injured $Atg8a^{\Delta4}$ flies by expressing 3xmCherry-tagged Atg8a under the control of the Atg8a promoter (Fig. 1b, c). Contralateral uninjured wings showed no obvious axon morphology defects (Fig. S1b). Flies null mutant for Atg5 ($Atg5^{5cc5}$) or trans-heterozygous with another allele ($Atg5^{5cc5}/Atg5^{CR}$) also failed to properly clear debris at 5 dpi, and this defect could be rescued as well by an endogenous promoter-driven Atg5 transgene (Fig. 1d, e).

We recently generated and characterized flies expressing a lipid conjugation-deficient version of Atg8a, where we replaced the codon coding for the glycine near the C-terminus (Gly116, to be conjugated to PE) with a stop codon in the endogenous Atg8a locus ($Atg8a^{G116*}$). These mutant flies are thus defective in Atg8a lipidation[38]. $Atg8a^{G116*}$ animals also presented defective axon debris clearance (Figs. 2a, c and S2), phenocopying $Atg8a^{\Delta4}$ and conjugation system mutants. These results demonstrate that the Atg8a conjugation system and Atg8a lipidation itself are required for proper clearance of axon debris after injury.

Nutrient limitation and other autophagy activating signals are relayed through the upstream-acting Atg1 kinase complex. We thus selected two viable mutants for genes encoding Atg1 complex members: an Atg101 null ($Atg101^{\Delta6h}$) and an Atg17/FIP200 severe hypomorph allelic combination ($Atg17^{\Delta130}/Atg17^{MI}$, note that Atg17 null mutants are pupal lethal so unsuitable for wing injury analysis) and gauged whether debris clearance depends on Atg1 kinase complex activity. To our surprise, neither of these mutants accumulated undegraded axon fragments (Figs. 2a–c and S2). Importantly, both mutants abrogated the autophagic process similarly to the conjugation system mutant $Atg5^{5cc5}$ as evidenced by severe depletion of punctate 3xmCherry-Atg8a in uninjured animals in the same wing area as used for axon imaging (Fig. 2d, e). This indicates that a process relying on the conjugation system but not requiring the Atg1 complex is at play during debris engulfment.

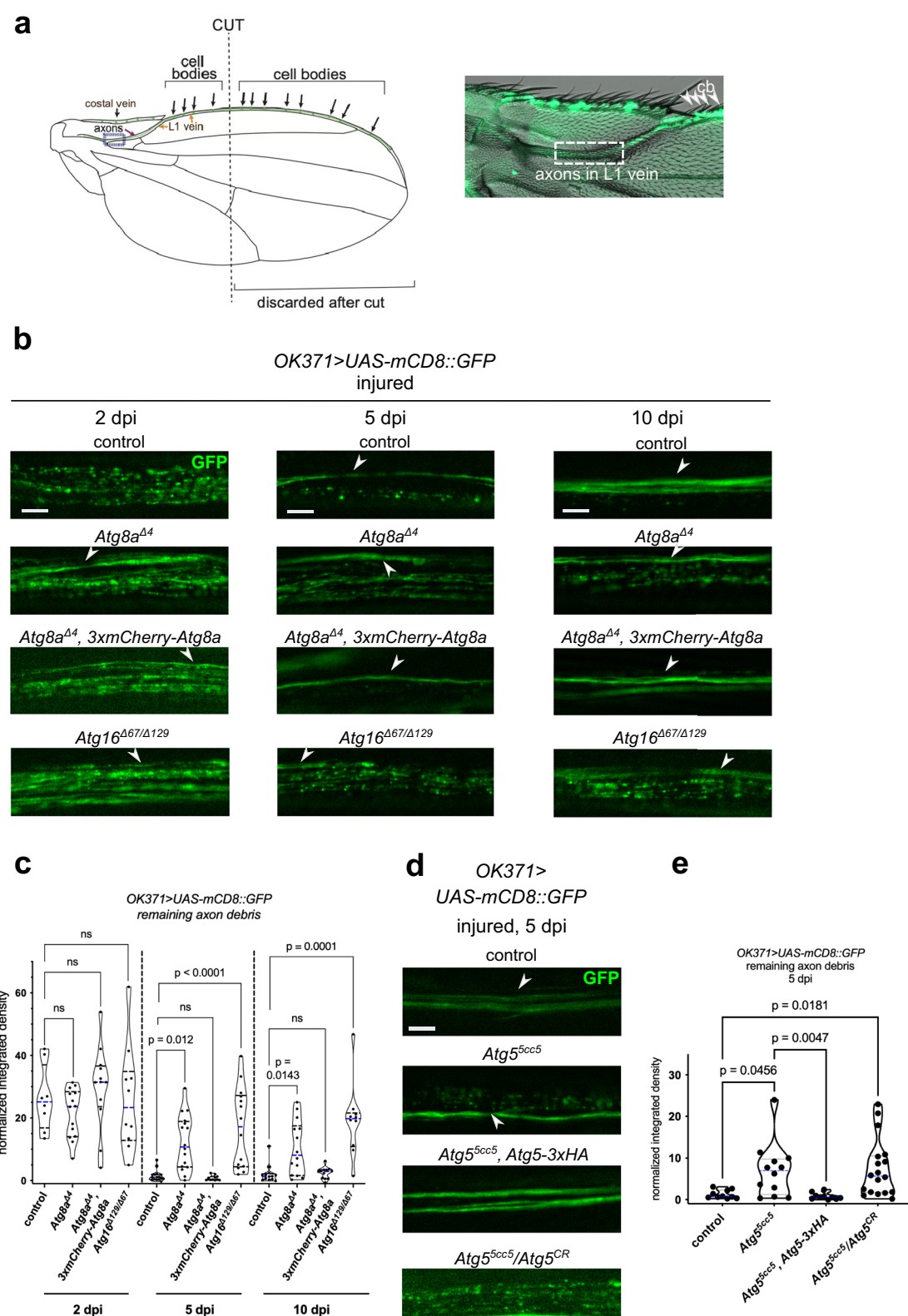

The Atg1 complex is dispensable for LAP, so we hypothesized that LAP could mediate the processing of internalized axon debris in glia. First, we tested this idea by disruption of a specific domain of Atg16 that is important for LAP but dispensable for canonical autophagy. Only the N-terminal half of Atg16 is essential for autophagy, while the C-terminal WD40 repeats are also necessary for LAP, both in cultured cells and in vivo[40,41]. We previously characterized a MiMIC transposon insertion (Atg16[MI]) truncating the WD40 repeats of *Drosophila* Atg16

protein, which does not disrupt autophagy[42]. Indeed, homozygous Atg16[MI] flies and flies trans-heterozygous for Atg16[MI] and the Atg16[Δ67] null mutant allele showed debris clearance defects, further supporting a role for LAP but not for canonical autophagy in debris engulfment (Figs. 2f, g and S2).

Next, we inquired whether Atg8a lipidation is activated after neural injury. As wing cellular material is scarce, we subjected flies to repetitive traumatic brain injury (TBI)[43] and confirmed that this

**Fig. 1 | The Atg8a conjugation system participates in axon debris clearance in the wing nerve. a** Schematic of the fly wing injury model and a combined DIC-fluorescent image of the L1 wing vein nerve with GFP-labeled axons. Left panel illustrates neuronal cell bodies that project axons through the L1 vein, imaged in the region of the arch near the blue dashed rectangle. Cutting the wing in half severs most axons, which degenerate and are cleared during the following days. See also Fig. S1a. Right panel shows the anatomy of the labeled nerve and cell bodies (cb) (green) with respect to external wing morphology. White dashed rectangle shows the region used for imaging. **b** Time course analysis of axon debris removal in the *Drosophila* wing nerve. Single optical slices of fragmented nerve bundles are shown for the indicated genotypes on *OK371> UAS-mCD8::GFP/+* background at 2, 5 and 10 days post wing injury (dpi). **d** Single optical slices of degenerating wing nerves of the indicated genotypes on *OK371> UAS-mCD8::GFP/+* background at 5 dpi. For the *Atg5cc5/AtgSCR00038* genotype, female flies were used. **c, e** Quantification of axon debris abundance in single-slice images of genotypes in (**b**) and (**d**), respectively. Truncated violin plots with median and quartiles are shown. Independent experiments are separated by a dashed line. **c** One-way ANOVA with Šídák's multiple comparisons test (2 dpi dataset, *p* = 0.4120, ns not significant, *p* values in order of appearance: >0.9999, >0.9999, >0.9999, respectively) and Kruskal–Wallis test with Dunn's multiple comparisons test (for the 5 dpi [*p* < 0.0001] and 10 dpi [*p* = 0.0001] datasets) was used for statistical analysis, respectively. *n* = 8, 14, 12, 10, 16, 16, 11, 14, 13, 14, 12, 10 biologically independent animals. ns not significant, control vs. *Atg8aΔ4, 3xmCherry-Atg8a* both at 5 dpi and 10 dpi: *p* > 0.9999. **e** Kruskal–Wallis test with Dunn's multiple comparisons test (*p* = 0.0004). *n* = 10, 12,10,18 biologically independent animals. Uninjured axons in injured L1 veins are denoted with arrowheads. Scale bar: 5 µm. *p* values indicated in (**c**) and (**e**) are corrected for multiple comparisons. Source data are provided as a Source Data file.

induces axon fragmentation in the brain (Fig. S3a). This was accompanied by a change from diffuse to punctate localization of the endogenous Draper protein (Fig. S3b), which reflects its translocation from plasma membrane to glial phagosomes[15]. Punctate localization of endogenous Atg8a was also evident 1 day after TBI, which depends on its lipidation as Atg8a protein distribution was diffuse cytosolic in the lipidation mutant *Atg8aG116** (Fig. S3c). In western blots of brain extracts, we found that in line with previous observations[43], the abundance of the lipidated form of Atg8a (Atg8a-II) increased upon brain injury (Fig. 2h). Levels of Atg5, the Atg12-Atg5 conjugate and Atg16 did not change after TBI (Fig. S3d, e). We next evaluated glial-specific responses in vesicle-associated Atg8a by expressing mCherry-Atg8a in glia and performing wing injury, which indeed caused an increase in Atg8a puncta number (Fig. S3f). Since the selective autophagy cargo Ref(2)P/p62 level did not change in glia after injury (Fig. S3g), these data are compatible with activation of LAP, rather than autophagy.

LAP is initiated by receptor-mediated internalization of extracellular ligands such as apoptotic cells, immune complexes, and pathogens. In *drpr* homozygous null mutant flies, glial cells fail to even contact axon debris[8,10]. Importantly, heterozygosity for *drpr* is enough to strongly interfere with axon debris phagocytosis[44], so we decided to use this sensitized genetic background to test if halving the level of Draper has any further effect on axon debris clearance in LAP mutants. We thus combined a heterozygous *drpr* null allele (*drprΔ5*) with various *Atg* gene mutants and quantified axon fragments after nerve cut. Heterozygosity for *drprΔ5* never enhanced the debris clearance defect seen in *Atg8aΔ4, Atg8aG116** or *Atg5cc5* mutant flies (Fig. S4a–c). As expected, *drpr* heterozygosity led to accumulation of axon debris in *Atg101Δ6h*, where autophagy is disrupted but LAP is functional (Fig. S4e). Interestingly, we saw fewer remaining axon fragments upon *drpr* heterozygosity in case of *Atg8aΔ4* compared to single mutants, in which the main Atg8a protein isoform is missing[39]. Since the *Atg8a* lipidation mutant and conjugation-defective animals do not show fewer axon fragments, these suggest another lipidation-independent role for Atg8a, in addition to its previously published such functions in other contexts[38,45].

## LAP in glia drives debris elimination

Neuronal knockdown of *Atg12* and *Atg17* causes a transient delay in axon degeneration in the injured wing at 1 dpi, but this effect is already gone by 3 dpi[46]. Based on this and our data, we sought to corroborate the specific requirements for autophagic complexes in debris engulfment by glia. We drove RNAi against *Atg5*, *Atg16* and *Atg1* in glia with *repo-Gal4* using dsRNAs targeting two independent regions for both *Atg5* and *Atg16*. RNAi knockdown of an unrelated control gene, *white* (*w*), was used as a negative control. Wing axons were labeled by an independent binary expression system, the Q system, using *VGlut-QF, QUAS-mCD8::GFP*. Debris removal defects were evident in glial *Atg5* and *Atg16* knockdown experiments but not for *Atg1* silencing at 5 dpi

(Figs. 3a–f and S5a–c). To functionally test the efficiency of *Atg* RNAi knockdowns, we utilized a classical autophagy assay where accumulation of the selective autophagy cargo Ref(2)P/p62 mirrors the degree of impediment in its autophagic degradation[47]. We expressed GFP-Ref(2)P together with these RNAi constructs and imaged GFP-Ref(2)P puncta in wing glia. All *Atg* knockdowns including *Atg1* dramatically accumulated Ref(2)P (Fig. 3g, h). We validated mRNA knockdowns by qRT-PCR (Fig. S6). We also silenced *Atg8a* in glia with two independent RNAi constructs specifically during adulthood using the drug-inducible, Gal4-based GeneSwitch system to avoid potential developmental consequences of *Atg8a* protein loss[48]. *Atg8a* knockdown in adult glia led to debris accumulation (Figs. 3i, j and S5e), similar to the down-regulation of conjugation system components. Since silencing of the gene coding for the autophagosomal Qa SNARE Syntaxin 17 (Syx17) did not impair debris elimination (Figs. 3k, l and S5d), even though it blocks autophagosome-lysosome fusion in *Drosophila* tissues[49], this also supports our model that canonical autophagy acting through Syx17 is dispensable for debris clearance. No gross morphological changes were apparent in uninjured wing nerves in either the density or spacing of glia or their interaction with axons upon *Atg16* knockdown, arguing for a specific defect in debris clearance rather than a developmental abnormality (Fig. S5h). In line with the above data, glial knockdown of the *Atg13-* and *Atg17/FIP200*-encoded subunits of the Atg1 complex had no effect on axon debris persistence (Fig. S7a, b), despite causing a significant increase in the number of GFP-Ref(2)P puncta and their specific knockdown measured by qRT-PCR (Figs. S6 and S7c, d). Thus, autophagy in wing glia is dispensable for axon debris removal.

## LAP in glia promotes phagosome maturation after engulfment

Based on these observations, we postulated that LAP is at play during debris processing after phagocytosis. During LAP, early phagosomes mature into special late phagosomes (LAPosomes) through phosphatidylinositol-3-phosphate synthesis, and LC3 conjugation to the single-membrane that finally culminate in fusion of the phagosome with the lysosome[25]. In the absence of LC3 conjugation, phagolysosome formation is abrogated. We observed axon debris within glial membrane-derived phagosomes after injury in wing nerves (Fig. S7e). We then wondered which step of phagosome maturation is affected upon inhibition of LAP. We readily observed the colocalization of axon fragments with glial Rab5-positive early phagosomes, glial Rab7-positive late phagosomes as well as glial Lamp1-positive lysosomes at 3 dpi (Figs. 4a and S8a, b). Knockdown of *Atg16* in glia did not affect axon fragment uptake into Rab5and Rab7vesicles in glia, but it caused a statistically significant decrease in axon debris delivery to lysosomes (Figs. 4a, b and S8a, b). This is consistent with our model that Atg8a conjugation to debris-containing phagosomes is necessary for their efficient fusion with lysosomes.

To gain more insight into the nature of axon fragment phagocytosis during LAP, we performed transmission electron microscopy

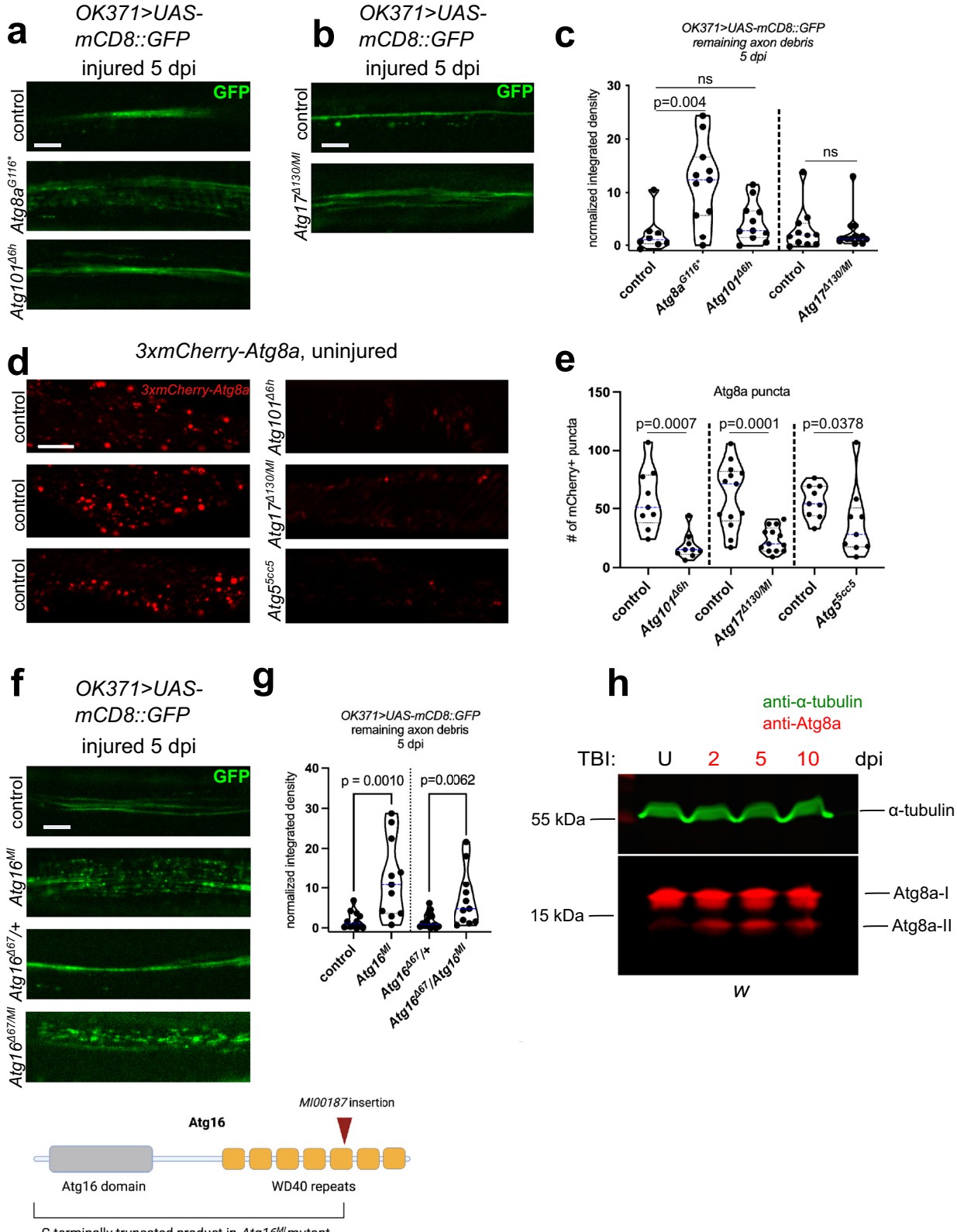

(TEM) of wing nerve cross-sections. The ultrastructure of the uninjured nerve showed tightly packed axons surrounded by glial protrusions (Fig. 5a), as described previously[50,51]. At 2 dpi, a few uninjured axons are still seen in both control and glial *Atg5* knockdown wing nerves (Fig. 5b, c), including the giant axon of the dTSM sensory neuron whose cell body is located proximally to the wing cut site (at the wing margin-arch boundary of the L1 vein, see Figs. 1a and S1a). Most axons, however, fall apart into small fragments, which become more electron-dense due to

Wallerian degeneration that is accompanied by degradation of proteins including microtubules by calpains[52,53] in both genotypes (Fig. 5b, c). Glial cells are easily recognized because their cytoplasm contains many electron dense free ribosomes that are lacking in axons. Glial cells appear to contain phagocytosed axon fragments, which are surrounded by the inner axon-derived and outer glial phagosome membranes (Fig. 5b, c). Note that if these phagocytosed axon fragments were captured into double-membrane autophagosomes, then vesicles

**Fig. 2 | The Atg1 kinase complex is dispensable for axon debris elimination, while Atg8a lipidation and a functional Atg16 WD40 domain is required for this process. a, b, f** Single optical slices of degenerating wing nerves of the indicated genotypes on *OK371> UAS-mCD8::GFP/+* background at 5 days post wing injury (dpi). Scale bar: 5 μm. **c, g** Quantification of axon debris abundance in single-slice images of the indicated genotypes at 5 dpi. **c** Kruskal–Wallis test with Dunn's multiple comparisons test (*Atg8a* and *Atg101* mutants, *p* = 0.0068) and unpaired, two-tailed Mann–Whitney test (*Atg17* mutant), *n* = 8, 11, 11, 11 and 13 biologically independent animals. *p* = 0.4266 for control vs. *Atg101⁴⁶ʰ*, *p* = 0.7762 for control vs. *Atg17ᴬ¹³⁰/ᴹᴵ*, ns not significant. **g** unpaired, two-tailed Mann–Whitney tests, *n* = 12, 11, 14, 11 biologically independent animals. **d** Single-slice images of *Atg8a promoter-3xmCherry-Atg8a*-expressing uninjured wing L1 veins of the indicated genotypes.

Scale bar: 10 μm. **e** Quantification of the number of 3xmCherry-Atg8a⁺ puncta in single-slice images of genotypes in (**d**). Unpaired, two-tailed *t*-test (*Atg101* and *Atg17/FIP200* mutants) and unpaired, two-tailed Mann–Whitney test (*Atg5* mutant) were used for statistics. *n* = 9, 9, 13, 13, 9 and 9 biologically independent animals. **h** Western blots of *w* head extracts from uninjured (U) and TBI-treated flies collected at 2, 5 and 10 days post-TBI (dpi). Endogenous Atg8a protein forms and α tubulin (loading control) are indicated. Atg8a-II is the lipid-conjugated form of Atg8a-I. Western blotting was independently repeated twice with similar results. **c, e, g** Independent experiments are separated by a dashed line. Truncated violin plots with median and quartiles are shown. *p* values indicated in (**c**) (left side) are corrected for multiple comparisons. Source data are provided as a Source Data file.

with 4 surrounding membranes would be present. The only ultrastructural difference that we saw between control and glial *Atg5* knockdown wing nerves was that injury-induced late phagosomes appeared less condensed in *Atg5* RNAi glial cells (Fig. 5c), which supports our model of LAP promoting late phagosome maturation and fusion with lysosomes.

Phagosome maturation during LAP depends on phosphatidylinositol 3-phosphate generated by the Vps34 complex on the phagosome membrane[25]. To dissect which of the Vps34 complexes support debris clearance by glia in flies, we depleted specific complex subunits as well as *Vps34* in glia and followed axon fragment persistence after nerve transection. At 5 dpi, knockdown of *Vps34*, *UVRAG* and *Rubicon* all increased the amount of remaining debris in wing nerves, whereas *Atg14* RNAi had no effect[32,54] (Figs. 6a–d and S5f, g). We confirmed *Atg14* and *Vps34* RNAi efficiency using the GFP-Ref(2)P accumulation assay for their ability to disrupt autophagy (Fig. S9a, b), as expected. qRT-PCR also validated target downregulation (Fig. S6). These results imply that axon debris clearance depends on the LAP-specific Vps34 complex in glia.

### Rubicon associates with debris-containing vesicles
Rubicon in mammals serves as the lynchpin of LAP-specific late phagosome (LAPosome) formation; besides its role in stimulating Vps34 activity on the phagosome, it stabilizes the NOX2 complex that generates ROS on LAPosomes, facilitating LC3-II incorporation[55,56]. Little is known about the fly ortholog of *Rubicon*[57,58], so we sought additional evidence for its involvement in glial debris phagocytosis. We created frameshift mutants of *Rubicon* (*Rubiconᶠˢ¹* and *Rubiconᶠˢ²*) by introducing small indels into the *Rubicon* coding region using CRISPR-Cas9 (Fig. S9c). Interestingly, *Rubicon* mRNA level halved in the *Rubiconᶠˢ¹* mutant presumably due to nonsense-mediated decay (Fig. S9d). Similar to *Rubicon* RNAi, *Rubiconᶠˢ¹* hemizygotes and *Rubiconᶠˢ¹/ᶠˢ²* transheterozygotes failed to clear axon debris in injured nerves (Figs. 6e–h and S2). We further validated this with *Rubiconᶠ⁰⁴⁴⁶²*, an independent, hypomorphic mutant allele that also led to a failure of removing axon fragments after injury (Fig. S9e, f). *Rubiconᶠˢ¹* failed to synergize with heterozygosity for *drpr^Δ5*, so it behaved similar to Atg8a conjugation system mutants (Fig. S4d). Finally, we utilized a tissue specific CRISPR knockout (tsKO) approach to disrupt *Rubicon* only in glia. We expressed *Cas9* in glia with *repo-Gal4*, expressed a control (*w*) vs. *Rubicon* single guide RNA and labeled axons with *VGlut-QF, QUAS-mCD8::GFP*. At 5 dpi, remaining debris was apparent in *Rubicon* but not in control tsKO wing nerves (Figs. 6i, j and S2).

Since Rubicon appears to be a key regulator of LAP, we overexpressed Rubicon in glia and assessed axon fragment density at a relatively early time point after wing injury. Interestingly, while axon fragments were still abundant in controls, glial Rubicon overexpression decreased the pool of undigested axon fragments, indicating that increasing Rubicon level is sufficient to boost debris clearance (Figs. 6k, l and S2).

How is Rubicon regulated after injury to stimulate LAP? We first hypothesized that Rubicon is upregulated upon wing injury. However,

we did not detect increased *Rubicon* mRNA levels in wings at 1 or 3 days after wing transection (Fig. S10a). Next, we followed Rubicon localization in glia. Rubicon is mainly found on vesicles in mammalian cells[59,60]. We expressed Rubicon::mRFP in glia and labeled axonal membranes with GFP. In wing nerve glia, Rubicon::mRFP showed a punctate distribution, which rarely overlapped with Rab5-positive early endosomes/phagosomes or Lamp1-marked lysosomes (Fig. S10b, c). Apparently, almost all Rubicon dots colocalized with Rab7-positive late endosomes/phagosomes in uninjured and injured wing nerve glia (Fig. 7a). The massive glial phagocytosis of axon debris that takes place after wing nerve injury caused a decrease in the number of Rubicon-Rab7 vesicles in glia, with a concomitant increase in their size (Fig. 7b). Interestingly, this coincided with a decrease in Rubicon-mRFP intensity on Rab7 vesicles (Fig. 7b). As expected, we observed colocalization of glial Rubicon::mRFP and GFP-labeled axon debris at 1 and 2 days after wing nerve injury, indicating that engulfed axon fragments are contained in LAPosomes (Fig. 7c, d).

### Atg8a is recruited to phagocytosed axon debris in glia
The key event of LAP is LC3/Atg8a recruitment to phagosomes. To ascertain that LAP promotes axon debris breakdown in glia, we labeled axon fragments with myr::GFP and expressed mCherry-Atg8a in glia. We indeed saw colocalization of punctate glial mCherry-Atg8a and GFP-labeled axon debris at 2 dpi (Fig. 8a). As GFP is quickly quenched in acidic lysosomes, axon-derived GFP-positive structures decorated with glial Atg8a correspond to LAPosomes. We next silenced *Vps34* to perturb the maturation of these LAPosomes. This caused an increase in overlap between glial mCherry-Atg8a and phagosomes containing GFP-labeled axon fragments (Fig. 8a, b). Our findings provide evidence for Atg8a and Rubicon recruitment to axon debris-containing phagosomes, lending further support to our genetic analyses that it is LAP that promotes lysosomal degradation of internalized axon debris in glia.

### LAP promotes survival after traumatic brain injury (TBI)
To establish the physiological relevance of LAP in glia, we again turned to the traumatic brain injury model. We exposed flies to brain injury on days 1, 3 and 6 to randomly induce mild damage to neurons. We then compared the survival of control and the LAP-specific mutant *Atg16ᴹᴵ* homozygous flies with or without injury. *Atg16ᴹᴵ* mutants displayed increased mortality after TBI (Fig. 9a). Importantly, the *Atg16ᴹᴵ* mutation did not affect the survival of uninjured flies (Fig. 9b). Thus, efficient debris clearance by LAP may contribute to survival after TBI.

## Discussion
We have now established that a subset of autophagy genes is required in glia for proper removal of axon debris generated after nerve injury. Dispensability of the Atg1 complex and dependence on Vps34 complex members Rubicon and UVRAG indicate a role for LC3-associated phagocytosis in glia (Fig. 10). This is further strengthened by our *Atg16* WD40 mutant data, which does not abrogate canonical autophagy but impairs debris clearance. Such LAP-specific mutants and

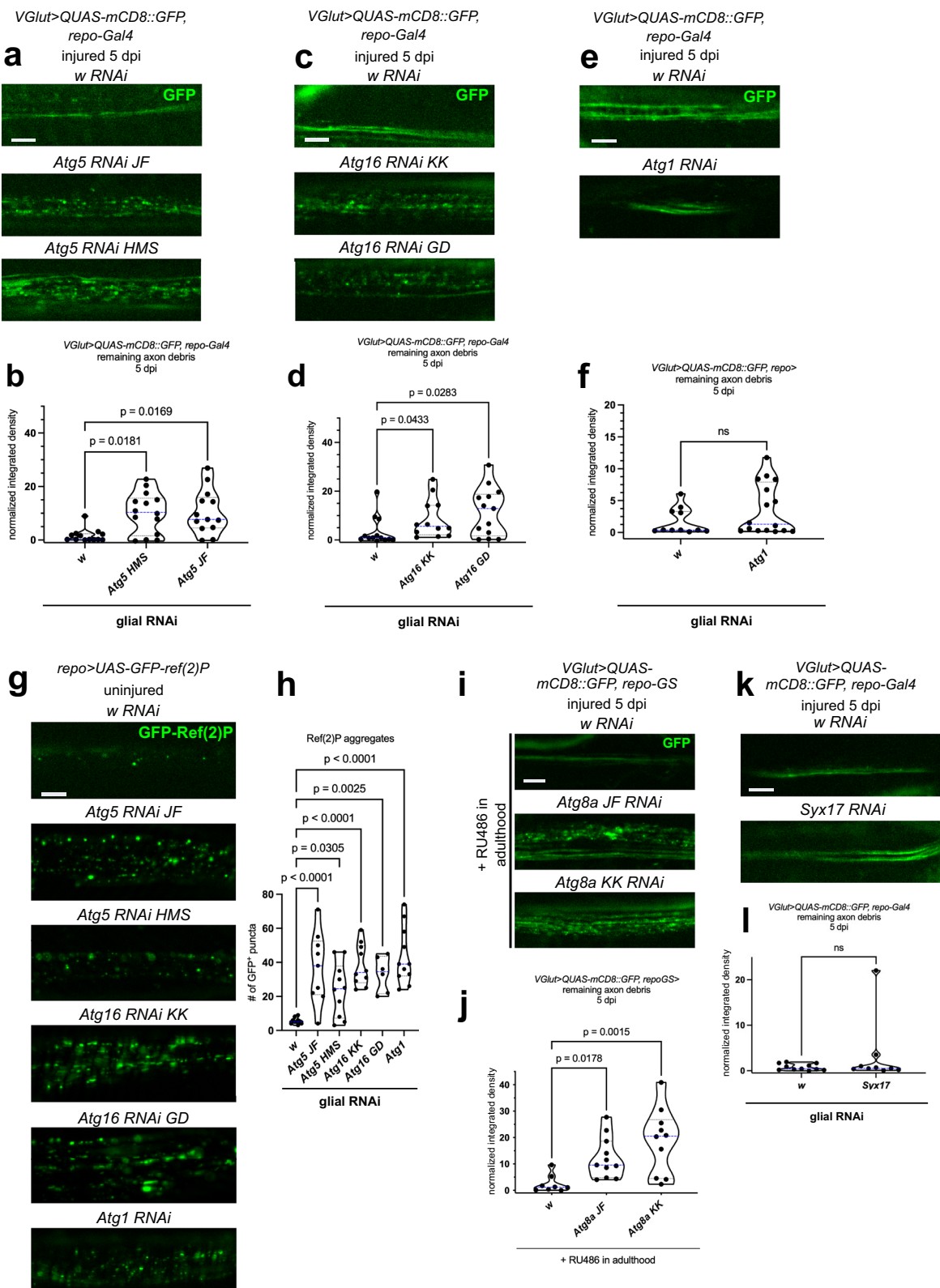

ultrastructural analysis exclude the possibility of autophagosomes being involved in the fusion between phagosomes and lysosomes, which was reported during *C. elegans* apoptotic cell corpse clearance[32,33]. LAP has mostly been characterized in macrophages during pathogen phagocytosis and efferocytosis. Although glial cells are efficient phagocytes, hardly any study has addressed the function of LAP in either microglia, astrocytes or in invertebrate glia[28–30,61]. The very recent demonstration that LAP in astrocytes promotes uptake of

microglial debris is an elegant but so far unique example of how LAP in glia promotes degradation of a physiologically relevant cargo in vivo[28].

We see two main questions related to LAP. The first one deals with the physiological occurrence and relevance of this process, which we focus on in this work. We have discovered an unexpected role for glial LAP in the breakdown of injured axons. Our genetic analyses in the *Drosophila* wing nerve perfectly match the data from previous publications in that all factors important for LAP proved to be necessary in

**Fig. 3 | A glial Atg8a conjugation system is required for efficient debris clearance. a, c, e, k** Single optical slices of degenerating wing nerves expressing the indicated RNAi-s in glia, driven by *repo-Gal4*, on *VGlut> QUAS-mCD8::GFP/+* background at 5 days post wing injury (dpi). **b, d, f, l** Quantification of axon debris abundance in single-slice images of the indicated genotypes in (**a**), (**c**), (**e**) and (**k**), respectively. **b, d** Kruskal−Wallis test with Dunn's multiple comparisons test. **b:** $p = 0.0105$, $n = 13, 14, 13$ biologically independent animals, **d:** $p = 0.0224$, $n = 13, 12, 13$ biologically independent animals. **f** unpaired, two-tailed Mann−Whitney test, $n = 11, 16$ biologically independent animals. $p = 0.3115$, ns not significant. **l** unpaired, two-tailed Mann−Whitney test, $n = 11, 9$ biologically independent animals. $p = 0.7103$, ns not significant. **g** Single-slice images of uninjured wing L1 vein glia expressing *repo-Gal4*-driven *UAS-GFP-ref(2)P* and co-expressing the indicated

RNAi-s. **h** Quantification of the number of GFP-Ref(2)P puncta in single-slice images of genotypes in (**g**), one-way ANOVA with Šídák's multiple comparisons test. $p < 0.0001$, $n = 10, 9, 10, 10, 6, 11$ biologically independent animals. **i** Single optical slice images of degenerating wing nerves expressing the indicated RNAi-s in glia, driven by *repo-GeneSwitch (GS)*, on *VGlut> QUAS-mCD8::GFP/+* background at 5 dpi. Gal4 was activated only during adulthood with RU486. **j** Quantification of axon debris abundance in single-slice images of the indicated genotypes in (**i**), Kruskal−Wallis test with Dunn's multiple comparisons test, $p = 0.0023$, $n = 8, 11, 10$ biologically independent animals. *w* RNAi serves as negative control. Truncated violin plots with median and quartiles are shown. Scale bar: 5 µm. $p$ values indicated in (**b**), (**d**), (**h**) and (**j**) are corrected for multiple comparisons. Source data are provided as a Source Data file.

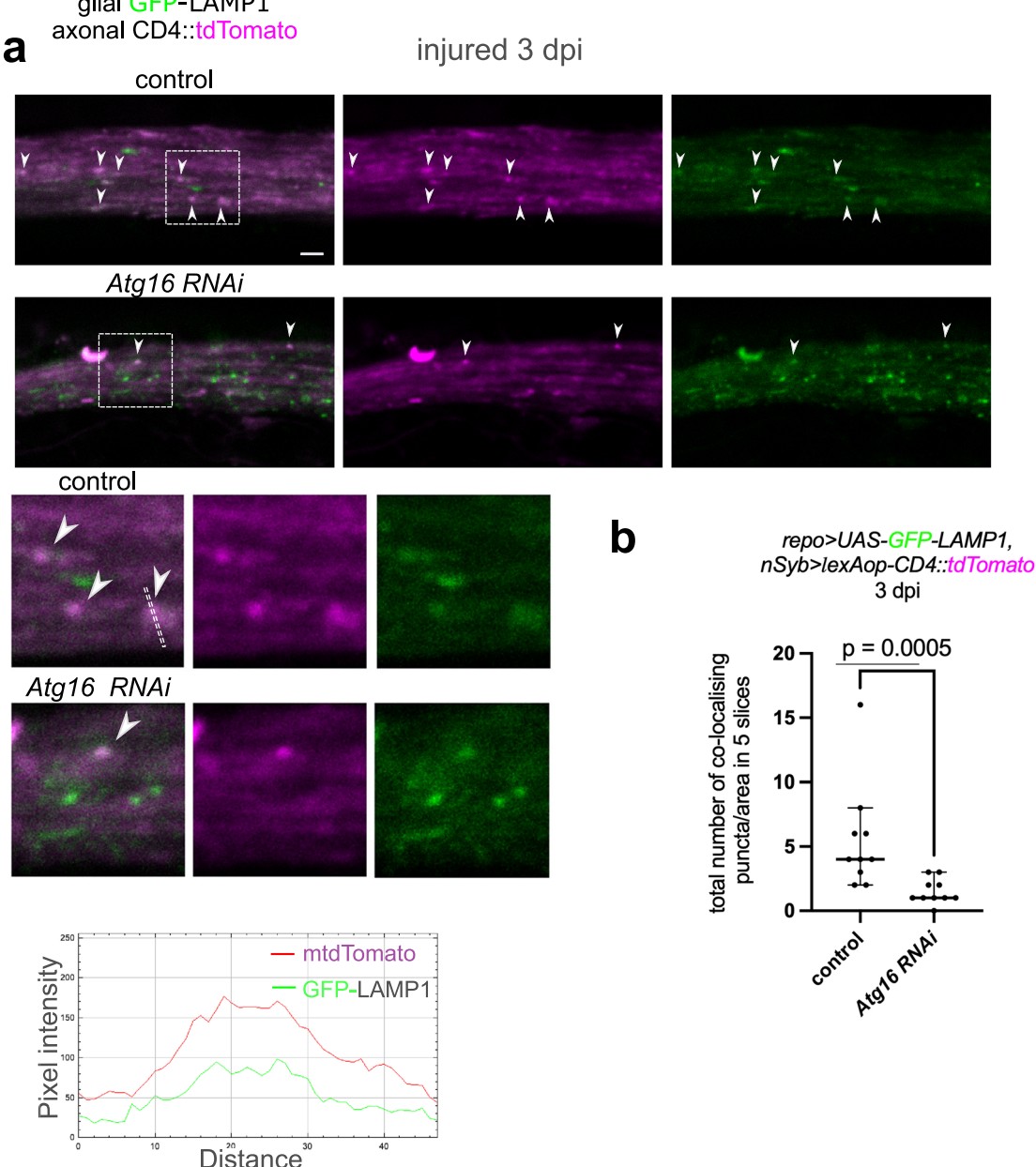

**Fig. 4 | Atg8a conjugation to phagosome membranes promotes phagosome-lysosome fusion. a** Confocal single-slice images of injured wing nerves at 3 days post wing injury (dpi) with or without the expression of *Atg16* RNAi in glia with *repo-Gal4*. Glial lysosomes are labeled with GFP-LAMP1, and axons with membrane-targeted tdTomato (*repo-Gal4> UAS-GFP-LAMP1,nSyb> lexAop-CD4::tdTomato*). Magnified images of the areas outlined by the dashed rectangles are shown below. Dashed line was used for intensity profile generation, pixel intensity plots over the indicated distance are shown below for each channel to demonstrate

colocalization. Arrowheads point to co-localizing puncta. Scale bar: 2 µm. **b** Quantification of the number of different co-localizing GFP-LAMP1 and CD4::tdTomato puncta in 5 consecutive single-slice images of genotypes in (**a**). Co-localizing puncta were counted in randomly selected 40 × 12 µm areas. Statistical analysis was performed with unpaired, two-tailed Mann−Whitney test. The graph shows the median with 95% confidence intervals. $n = 10, 10$ biologically independent animals. Source data are provided as a Source Data file.

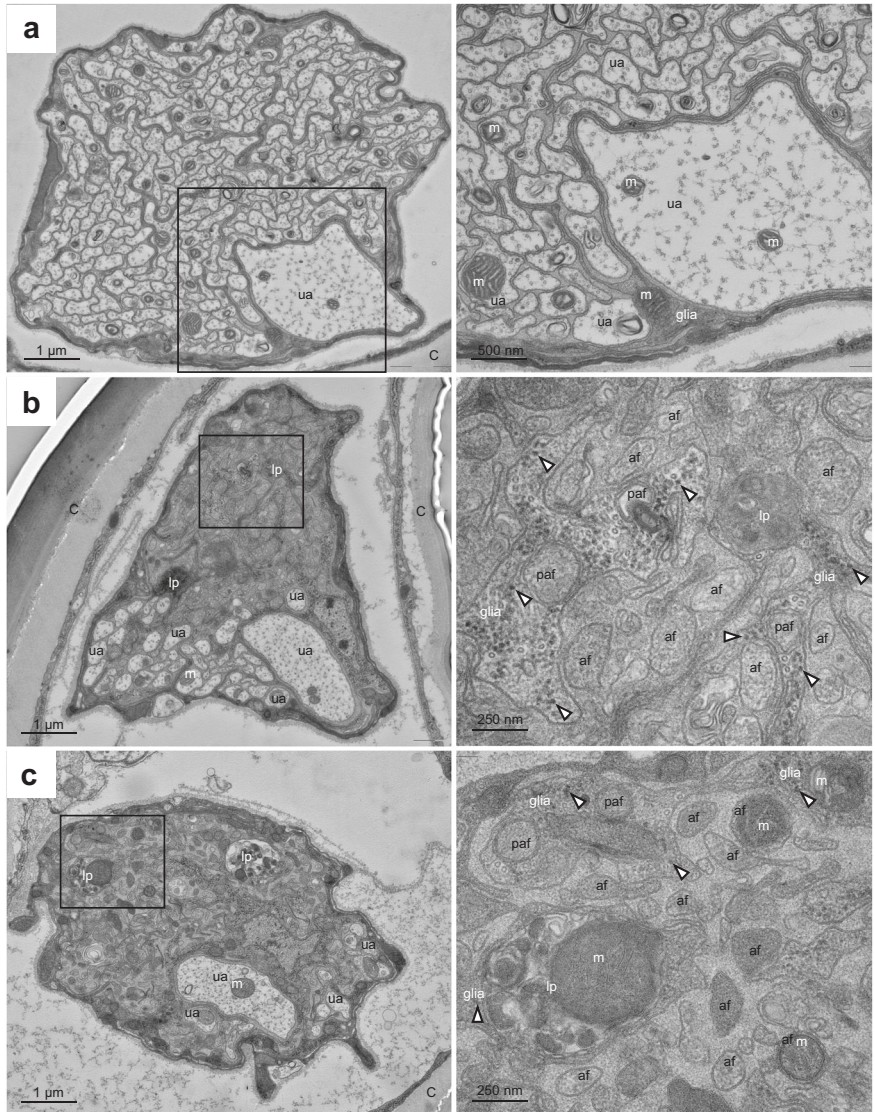

**Fig. 5 | Ultrastructure of uninjured and injured wing nerves. a** Control, uninjured wing nerve shows cross-sections of about 200 electron translucent uninjured axons (ua), in which mitochondria (m) are occasionally seen, and glial cells whose cytosol is more electron dense and contains various organelles including mitochondria (m). Right panel shows an enlarged view of the boxed area from the left panel. **b** 2 days after injury of control wings, uninjured axons (ua) are still present, but most of the wing nerve cross-section area is occupied by injured axon fragments and glia. The enlarged view of the boxed area (right panel) shows the ultrastructure of axon fragments (af, surrounded by a single axon-derived membrane) in the extracellular matrix (homogenous substance filling the spaces between glia and axon fragments). Glial cells are easily recognized based the presence of free ribosomes (some of which are indicated by arrowheads) in their cytoplasm, which are missing from uninjured and injured axons. Glial cells contain multiple potential phagocytosed axon fragments (paf), which have both the axon-derived membrane and the phagosome membrane, and their inner contents look like that of extracellular axon fragments. Late phagosomes (lp) that contain condensing engulfed material are also seen within glia. **c** 2 days after injury of wings with glial *Atg5* knockdown (*repo>Atg5* RNAi), uninjured axons (ua) are also present, and again injured axon fragments and glia occupy most of the cross-section area. The ultrastructure of the boxed area (enlarged in the right panel) is mostly similar to the injured control wing nerve: both the axon fragments (af) in the extracellular matrix and potential phagocytosed axon fragments (paf) within glia (with arrowheads pointing to some of the ribosomes in glial cells) are visible. Late phagosomes (lp) containing phagocytosed material are readily recognized within *Atg5* knockdown glia. Interestingly, the condensation of engulfed axon fragments seems delayed based on the presence of electron translucent areas and an undegraded large mitochondrion (m, compare its structure with those seen in axon fragments) within the late phagosome (lp) shown in the enlarged right panel. C cuticle covering the wing. *n* = 3, 3, 3 biologically independent animals.

glia for proper elimination of phagocytosed axon fragments including Rubicon, Atg8a lipidation and the Atg16 WD40 domain[25,27,41,56]. Such a role for LAP is further confirmed by the presence of phagocytosed axon fragments within glial LAPosomes based on colocalizations and ultrastructural data. Thus, the wing nerve injury paradigm is a perfect in vivo model to study LAP in glia. But what is the physiological relevance of LAP? This is not clear from the wing injury model, because the severed half of the wing is discarded, and this does not affect the survival of laboratory animals kept in culture vials. Our traumatic brain injury experiments with LAP-specific Atg16 WD40 domain mutant

animals indicate that LAP in glia may contribute to survival after TBI, a process that triggers acute neurodegeneration.

The second main question that we see deals with the molecular events during LAP. Little is known about the differences between endosome and phagosome maturation. The first steps of these processes are clearly different: phagocytosis of microbes, dead cells, and cell fragments utilizes specialized phagocytic receptors, while during endocytic uptake, smaller particles, macromolecular assemblies, and ligand-receptor complexes are internalized. Rubicon and LC3/Atg8 conjugation to single membranes have been shown to promote a

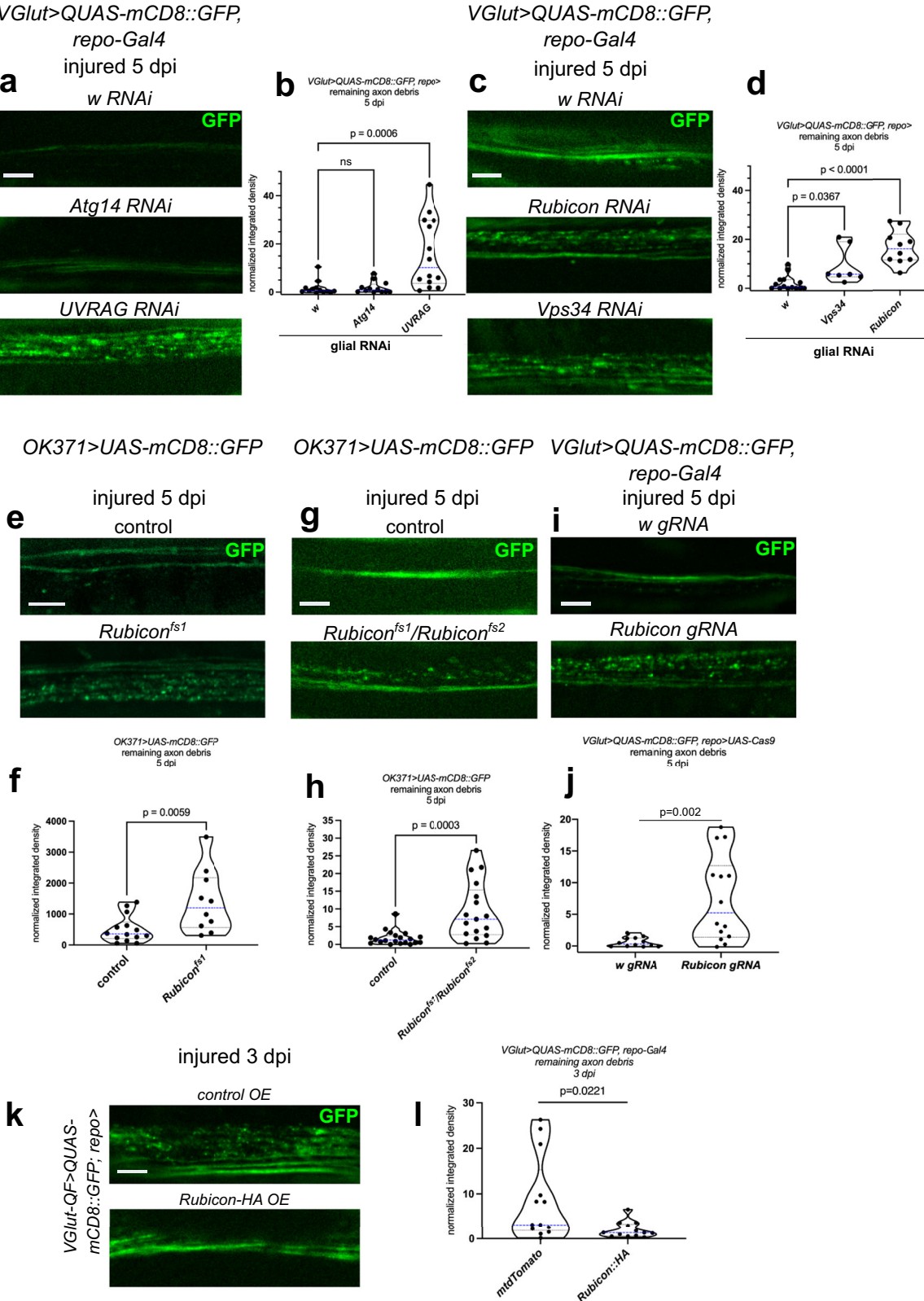

subset of these cases, with several examples for phagocytosis (LAP)[27,62]. Since phagocytic receptors such as Draper are internalized along with cell debris, these presumably collide with LAP components including Rubicon during the fusion/maturation of early phagosomes with Rab7-positive late phagosomes/endosomes. The C-terminal RH domain of Rubicon directly binds to Rab7[63], which explains its presence on Rab7+ vesicles. It is plausible to assume that Rubicon binds to the UVRAG-containing Vps34 complex on these vesicles. Rubicon likely interferes with endosome maturation (as described previously) by binding to and inhibiting these factors[59,60,64,65]. When the proper cargo arrives together with its phagocytic receptor such as Draper, this may somehow signal to Rubicon to promote phagosome maturation and cargo breakdown. After a decade of research into the role of Rubicon in LAP (mostly in cultured cells that are best suited to quick molecular pathway analysis), further studies are still necessary for uncovering the molecular details of this process[27]. Of note, a role for LC3, Rubicon and

**Fig. 6 | Rubicon, UVRAG and Vps34 promote LAP of axon debris. a, c** Single optical slices of degenerating wing nerves at 5 days post wing injury (dpi) expressing the indicated RNAi-s in glia, driven by *repo-Gal4* on *VGlut> QUAS-mCD8::GFP/+* background. **b, d** Quantification of axon debris abundance in single-slice images of the indicated genotypes as in (**a**) and (**c**), at 5 dpi. **b** Kruskal–Wallis test with Dunn's multiple comparisons test. *p* = 0.0003, *n* = 13, 12, 14 biologically independent animals. ns not significant: *p* > 0.9999. **d** Kruskal–Wallis test with Dunn's multiple comparisons test. *p* < 0.0001, *n* = 14, 7, 10 biologically independent animals. Single optical slices of degenerating wing nerves at 5 dpi in either the *Rubicon^fs1* mutant (**e**), in a female trans-heterozygous *Rubicon^fs1/fs2* mutant (**g**) or upon expressing Rubicon single guide RNA universally and *UAS-Cas9* with *repo-Gal4* in glia (**i**). *OK371> UAS-mCD8::GFP* (**e, g**) and *VGlut> QUAS-mCD8::GFP* (**i**) are used for axon labeling. **f, h, j** Quantification of axon debris abundance in single-slice images of the indicated genotypes as in (**e, g, i**), respectively, at 5 dpi. Unpaired, two-tailed Mann–Whitney tests: *n* = 14, 10 biologically independent animals (**f**) and *n* = 20, 17 biologically independent animals (**h**) and unpaired, two-tailed *t*-test (**j**): *n* = 12 and 14 biologically independent animals, were used for statistics. **k** Single optical slices of degenerating wing nerves at 3 dpi, glial overexpression (OE) using *repo-Gal4* on a *VGlut>QUAS-mCD8::GFP/+* background. *UAS-mtdTomato* is used as control for Rubicon overexpression. **l** Quantification of axon debris abundance in single-slice images of the indicated genotypes as in (**k**) at 3 dpi, unpaired, two-tailed Mann–Whitney test. *n* = 13, 13 biologically independent animals. Scale bar: 5 μm. Truncated violin plots with median and quartiles are shown. *p* values indicated in (**b**) and (**d**) are corrected for multiple comparisons. Source data are provided as a Source Data file.

the WD40 domain of Atg16 was also described during recycling of putative amyloid β receptors to the cell surface, impairment of which results in increased plaque accumulation, inflammation, and behavioral defects[54,61].

In addition to its function in debris recognition and glial activation, Draper may promote engulfment through the GEF complex Crk/Mbc/dCed-12 to stimulate the pinching of the forming phagosome[15]. Hinting to a role for Drpr in a later step such as phagosome maturation by LAP, *drpr* knockdown in brain glia was reported to induce accumulation of Atg8a around engulfed cell corpses[66]. This effect resembles our results with persisting Atg8a on phagosomes upon silencing of *Vps34*, which is likely due to two waves of Vps34 activity that were described during early and late stages of phagosome maturation, and its role in PI3P generation in the first round was found to be redundant with a class II PI3K[67]. Since the *drpr* orthologs *CED-1* (C. elegans) and *MEGF10/JEDI* (mammals) show a conserved function in engulfment of dead cells, we expect that LAP also fulfils conserved roles in CNS debris elimination in other species. This is illustrated by a glia eats glia paradigm of LAP in astrocytes promoting breakdown of C4b-opsonized microglial debris in mice[28].

We predict that glial LAP would support the mammalian nervous system based on recent studies showing the importance of microglial non-canonical autophagy in disease settings. First, a process reminiscent of LAP was described in myelin debris clearance by microglia in a murine multiple sclerosis (MS) model[68]. Microglial deletion of *ATG7* but not *ULK1* (ortholog of *Atg1*) leads to defective myelin degradation and impaired recovery from experimental autoimmune encephalomyelitis. Second, correct developmental wiring of the CNS also depends on phagocytosis of pruned supernumerary synapses. Intriguingly, *ATG7*-deficient microglia are inefficient in synaptic pruning and therefore mice with microglial deletion of this gene display repetitive behaviors and impaired social interaction[69]. Synaptic pruning could accordingly be under the control of LAP as well.

A recent paper suggested that increased neuronal expression of Rubicon during aging decreases autophagy and limits lifespan in multiple models, while loss of Rubicon upregulated autophagy and led to longevity[57]. Since our data demonstrate the importance of Rubicon in removing damaged axon debris by glia after nervous system injury, its expression is likely important in animals that need to survive in the wild where they are exposed to various injuries, unlike in the case of the artificial environment where laboratory animals are maintained. Indeed, mutations in the human RUBCN gene are associated with a familial form of ataxia with impaired lysosomal degradation[70].

LAP has recently been characterized in multiple phagocytic cell types with studies emphasizing its modulatory role[71]. First discovered in macrophages, LAP was later identified in other immune cells and non-professional phagocytes[26]. LAP in retinal pigment epithelium mediates breakdown of photoreceptor outer segments (POS, a neural tissue) in a circadian rhythm[72,73]. Beyond its well-established role in defence against pathogens, it is still a largely unanswered question how important LAP is in vivo. Knockout of *Rubicon* in all cells or *Atg5*

in macrophages does not completely abrogate phagocytic breakdown but still has severe long-term consequences if mice are repeatedly injected with dead thymocytes, such as the development of autoinflammatory systemic lupus erythematosus (SLE)-like disease[74], although an independent study using a genetic model of lupus did not confirm this[75]. Rubicon knockout also leads to enhanced hepatic inflammation and fibrosis in response to liver injury[76]. Undegraded cellular material can transform phagocytes into a proinflammatory state[74] that could also happen in the nervous system. Neuroinflammation contributes to the long-term consequences of both traumatic brain injury and neurodegenerative diseases[77], conditions accompanied with many dying cells. Our results point to a role of LAP in the phagocytic arm of the debris clearance pathway to prevent uncontrolled inflammation in response to uncleared material in the nervous system. Our Atg16 WD40 mutant data indeed suggest that LAP promotes recovery from the consequences of TBI in *Drosophila*. Based on our initial findings with glial Rubicon overexpression, we also expect that enhancing LAP (perhaps in a more specific way) in the injured nervous system could facilitate the removal of dead brain cells and debris in acute and chronic neurodegeneration.

## Methods

### Ethics statement

The research presented here uses the species *Drosophila melanogaster* for which no ethical approval is required in the Biological Research Center, Institute of Genetics, ELKH or by Hungarian authorities. Maintenance of transgenic *Drosophila melanogaster* at the Institute is regulated by license No. SF/66/2014 of the Gene Technology Authority Registry, Ministry of Agriculture of the Hungarian Government.

### *Drosophila* rearing and stocks
Flies were maintained at 25 °C on a cornmeal-yeast-agar-dextrose medium with Nipagin as preservative. *Drosophila melanogaster* males 4–8 days after eclosion were used for the experiments except when stated otherwise. We preferred males over females due to long incubation times of injured animals during which female egg laying and resulting larval stirring of fly food would lead to enhanced mortality of adults as they stick to fly food. Four to 8 days old females were only used in experiments where trans-heterozygous combinations of X chromosomal mutations were studied, including Figs. 1d and S1b: *Atg5^cc5/Atg5^CR00038* and Figs. 6g and S2: *Rubicon^fs1/Rubicon^fs2*. Flies were sacrificed after carbon dioxide anesthesia by decapitation or submerging into pure ethanol. *w; OK371-Gal4, UAS-mCD8-GFP*[78] was a kind gift from H. Aberle (Institute of Functional Cell Morphology, Heinrich-Heine-University, Düsseldorf). *UAS-GFP-LAMP1*[79], *Atg8a^Δ4,39*, *Atg16^Δ67,80*, *Atg16^Δ129,80*, *Atg5^cc5,81*, *Atg8a^G116,38*, *Atg17/FIP200^Δ130,82*, *UAS-GFP-ref(2)P*[83], *Atg16-3xHA*[42,80] and *3xmCherry-Atg8a*[19] driven by their endogenous promoters were previously described. *Atg101^Δ6h* was a kind gift of Wanzhong Ge (Institute of Genetics, Zhejiang University School of Medicine, Hangzhou, China)[84]. The *UAS-WT dRubicon::HA* stock was described earlier[57,58] and was contributed kindly by Mari Suzuki (Tokyo

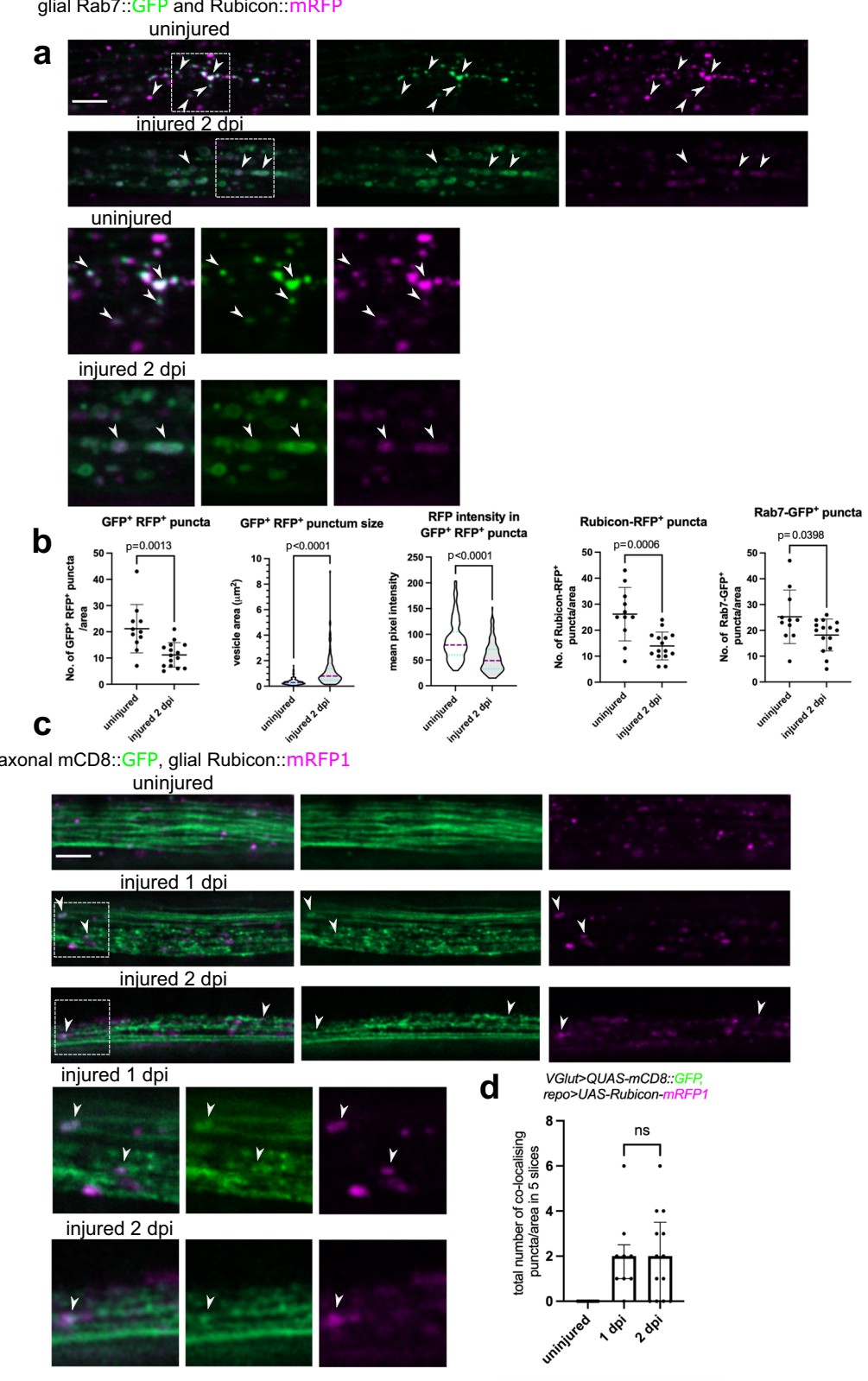

glial Rab7::GFP and Rubicon::mRFP

**a**

uninjured

injured 2 dpi

uninjured

injured 2 dpi

**b**

GFP⁺ RFP⁺ puncta

GFP⁺ RFP⁺ punctum size

RFP intensity in GFP⁺ RFP⁺ puncta

Rubicon-RFP⁺ puncta

Rab7-GFP⁺ puncta

**c**

axonal mCD8::GFP, glial Rubicon::mRFP1

uninjured

injured 1 dpi

injured 2 dpi

injured 1 dpi

injured 2 dpi

**d**

VGlut>QUAS-mCD8::GFP, repo>UAS-Rubicon-mRFP1

total number of co-localising puncta/area in 5 slices

Metropolitan Institute of Medical Science, Japan). The *repo-GeneSwitch* line was a gift of Véronique Monnier (Université de Paris, BFA Unit of Functional and Adaptative Biology, UMR 8251).

The following stocks were obtained from Bloomington *Drosophila* Stock Center: *repo-Gal4* (7415), *drpr^Δ5* (67033), *Mi{Trojan-QF2.2} VGlut^MI04979-TQF2.2* (60315), *QUAS-mCD8::GFP* (30002), *Atg17/FIP2OO^MI01469* (34198), *Atg16^MI00187* (30656), *Rubicon^f04462* (18773), *Atg5^CR00038* (81138),

*w^GL00094* (35573), *Atg14^HMS02025* (40858), *UVRAG^HMS01357* (34368), *Atg5^IF02703* (27551), *Atg5^HMS01244* (34899), *Atg1^GL00047* (35177), *Syx17^IF01937* (25896), *Atg8a^IF02895* (28989), *w sgRNA TKO.GS02468* (79543), *Rubicon sgRNA TKO.GS04756* (81781), *UASp-mCherry-Atg8a* (37750), *nSyb-lexA.DBD::QF.AD, 13x lexAop2-IVS-myr::GFP* (51954), *13xLexAop2-CD4::tdTomato* (77139), *UAS-IVS-myr::tdTomato* (32221), *UAS-Rab7::GFP* (42705), *UAS-GFP::Rab5* (43336), *tubP-Gal4* (5138) and *10XQUAS-*

**Fig. 7 | Glial Rubicon colocalizes with large Rab7-positive vesicles containing axon debris after injury. a** Confocal single-slice images of uninjured and injured wing nerves at 2 days post wing injury (dpi) expressing Rubicon::mRFP and Rab7::GFP in glia (*repo-Gal4> UAS-Rubicon::mRFP, UAS- Rab7::GFP*). **b** Single-slice quantification of puncta area, number, and mean pixel intensity for the indicated single and double positive vesicles as in (**a**). GFP⁺ RFP⁺ puncta, Rubicon-RFP⁺ puncta and Rab7-GFP⁺ puncta: unpaired, two-tailed *t*-test, *n* = 11 and 15 biologically independent animals, mean with standard deviation; GFP⁺ RFP⁺ punctum size and RFP intensity: two-tailed Mann–Whitney test, *n* = 233 and 162 puncta from *n* = 11 and 15 biologically independent animals, respectively. Truncated violin plots with median

and quartiles. **c** Confocal single-slice images of uninjured and injured wing nerves at 1 and 2 dpi expressing Rubicon::mRFP in glia. Axons are labeled with mCD8::GFP (*repo-Gal4> UAS-Rubicon::mRFP, VGlut> QUAS-mCD8::GFP*). **a, c** Magnified images of the areas outlined by the dashed rectangles are shown below. Arrowheads point to overlapping puncta. Scale bar: 5 µm. **d** Quantification of the number of different co-localizing Rubicon::mRFP and mCD8::GFP puncta in five consecutive single-slice images of genotypes in (**a**). Co-localizing puncta were counted in randomly selected 40 × 12 µm areas. *n* = 11, 13, 9 biologically independent animals, unpaired, two-tailed Mann–Whitney test. The median with interquartile range is shown. ns not significant, *p* = 0.9606. Source data are provided as a Source Data file.

6XGFP; Orco-2A-QF2 (92402). Knockdown of UVRAG by the transgenic UVRAG^HMS01357 RNAi construct[21,85] and Atg1 by the Atg1^GL00047 RNAi line[86,87] has been validated previously. We received the following stocks from Vienna *Drosophila* Resource Center: Atg8a^KK102155, Rubicon^KK108247, Vps34^KK107602, Atg13^KK100340, Atg17/FIP2OO^KK101847, Atg16^GD10140 (v2565l) and Atg16^KK102326. Standard meiotic recombination was used to generate composite transgenes on the same chromosome such as nSyb-lexA.DBD::QF.AD, lexAop2-CD4::tdTomato and Mi{Trojan-QF2.2}Vglut^MI04979-TQF22, QUAS-mCD8::GFP.

All TRiP lines were outcrossed before use to a *w* background (BDSC 5905) to remove the X chromosome, which contained a *scute* mutation in many of them. This mutation in a hemizygous state is presumably responsible for a severe reduction of the number of wing margin neurons that does not occur in outcrossed stocks.

To generate the Rubicon^fs1 and Rubicon^fs2 mutants, we crossed vas-Cas9 (VK00027, Bloomington # 51324) to a constitutively expressed single guide RNA stock (GS04756, Bloomington # 81781) that targets second coding exon of *Rubicon*. Single potential indel events were isolated and individual flies were tested for *Rubicon* indels by PCR, T7 Endonuclease I assay and sequencing. Frameshift mutants were then selected for further experiments.

To generate the genomic promoter-driven Atg5–3xHA transgene, we PCR-amplified a 9135 bp region containing the *Atg5* gene and its promoter using genomic fly DNA and primers 3′-TCTAGAGGTA CCAACATCCTTGTAGTCCACCGATAAATGC-5′ and 5′-GGCGCGCCAG CGGCCGCAGCGAGCACAGAGGCAAGTCG-3′ and cloned the product into the pGen-3xHA vector[20] as a NotI-Acc65I fragment. In this construct, the endogenous stop codon was replaced by 3xHA coding sequences. After sequencing the vector, the transgenic fly lines were established by Bestgene Inc.

To obtain the UAS-Rubicon::mRFP transgene, a Gateway entry vector (kind gift of Mari Suzuki, Tokyo Metropolitan Institute of Medical Science, Japan) bearing the full length *Drosophila Rubicon* cDNA coding sequence (*pENTR-WT dRubicon*)[58] was recombined with pTWR from the Carnegie *Drosophila* Gateway Vector Collection (Murphy lab) in an LR Clonase reaction according to the manufacturer's protocol (Gateway LR Clonase II Enzyme Mix, Thermo Fisher Scientific). Resulting clones were sequenced and used for *Drosophila* transgenesis in *w^1118* background based on standard protocols for P-element transformation.

For GeneSwitch (GS) experiments, progeny of *repo-GS* crossed with RNAi lines were raised in normal food during development, then 1–3-day old adult males were kept for 5 days on food supplemented with 25 µg/ml mifepristone (RU486) before axotomy and thereafter during the course of the experiment.

**RNA isolation and RT-qPCR**
For RNA isolation, 20 adult male carcasses or 40 wings were disrupted with a motor pestle for 2 × 1 min in 200 µl or 100 µl of TRIReagent, respectively and thereafter supplemented with 400 µl TRI Reagent. Total RNA was extracted with the Direct-zol RNA MiniPrep (for carcasses) or Microprep (for wings) (Zymo Research). DNase I digestion was also performed. One µg (carcasses) or 60 ng (wings) total RNA was reverse transcribed in 10 µl reaction volume using the RevertAid First

Strand cDNA Synthesis Kit (Thermo Scientific). qRT-PCR was performed in 20 µl reactions in technical triplicates using the PerfeCTa SYBR Green FastMix (Quantabio) with 1 µl cDNA and cycled on a Rotor-Gene Q qPCR machine (Qiagen) running Rotor-Gene software 2.3.1.49 (Qiagen), with the following program: 95 °C, 3 min; 45 cycles of 95 °C, 20 s, 58 °C, 20 s and 72 °C, 20 s followed by melting curve analysis. The data were normalized by the ΔΔCt method using *Ribosomal protein L32* (*RpL32*, also known as *rp49*) as an internal control. All primers were designed with Primer-BLAST (https://www.ncbi.nlm.nih.gov/tools/primer-blast) with amplicon length set to 80–150 bp and melting temperature set to 60 °C. One primer always spanned an exon-exon junction. Primers used for RT-qPCR are listed in Supplementary Table S1. Normalization to *RpL32* was used for qPCR datasets. All ΔCt-derived expression values were multiplied by the same scaling factor so that average of control values would equal 1 or 100.

**TBI and fly survival monitoring**
One to 3-day old flies were subjected to controlled shaking to induce traumatic brain injury as previously described[43]: cohorts of 10 flies in 2-ml tubes underwent 1 cycle of shaking at 4500 rpm for 10 s in a Precellys Evolution bead mill homogenizer (Bertin Technologies). Flies were allowed to recover for 10 min and transferred to food-containing vials laying on their side for incubation at 25 °C post-injury. After day 1, this treatment was repeated on days 3 and 6 for all 8 replicates. Uninjured controls were treated identically except the injury. Mortality was scored every 2–3 days while transferring flies to fresh vials. The experiment lasted until the death of the last injured fly.

**Western blotting**
One to 5 days-old flies were subjected to TBI as above, only cohorts of 10 flies in 2-ml tubes underwent 3 cycles of 10 s of shaking at 4500 rpm with 30 s dwelling time in between. Flies were allowed to recover for 10 min and transferred to food-containing vials for incubation at 25 °C for the indicated times post-injury. After freezing, heads were separated from bodies by sieving on dry ice. Four µl of 1x Laemmli sample buffer containing 100 mM DTT was added per head and heads were lysed with a motor pestle in a microfuge tube for 2x 1 min Extracts were boiled for 5 min at 100 °C and spun for 5 min 12,000 × g to remove debris. Supernatants (20 µl) were loaded on 15% SDS-polyacrylamide gels for Atg8a detection and on 10% gels for Atg5 and Atg16 blotting. After overnight blotting on PVDF, membranes were blocked with Intercept (TBS) Blocking Buffer (LI-COR) for 1 h at room temperature (RT). Primary antibodies were: anti-GABARAP/GABARAPL1/GABARAPL2 rabbit monoclonal (Abcam ab109364, clone number EPR4805) 1:2000 for Atg8a, anti-HA rabbit polyclonal (Sigma-Aldrich H6908) 1:1000 and anti-α-tubulin mouse monoclonal (Developmental Studies Hybridoma Bank AA4.3) 1:1000, diluted in Blocking Buffer and incubated for 1 h at RT. Secondary antibodies were anti-mouse IRDye 800CW (926-32210, LI-COR) and anti-rabbit IRDye 680RD (926-68071, LI-COR) 1:15,000, diluted in Blocking Buffer supplemented with 0.02% SDS and 0.2% Tween-20 and incubated for 1 h at RT. Blots were imaged and quantified in an Odyssey Clx instrument (LI-COR).

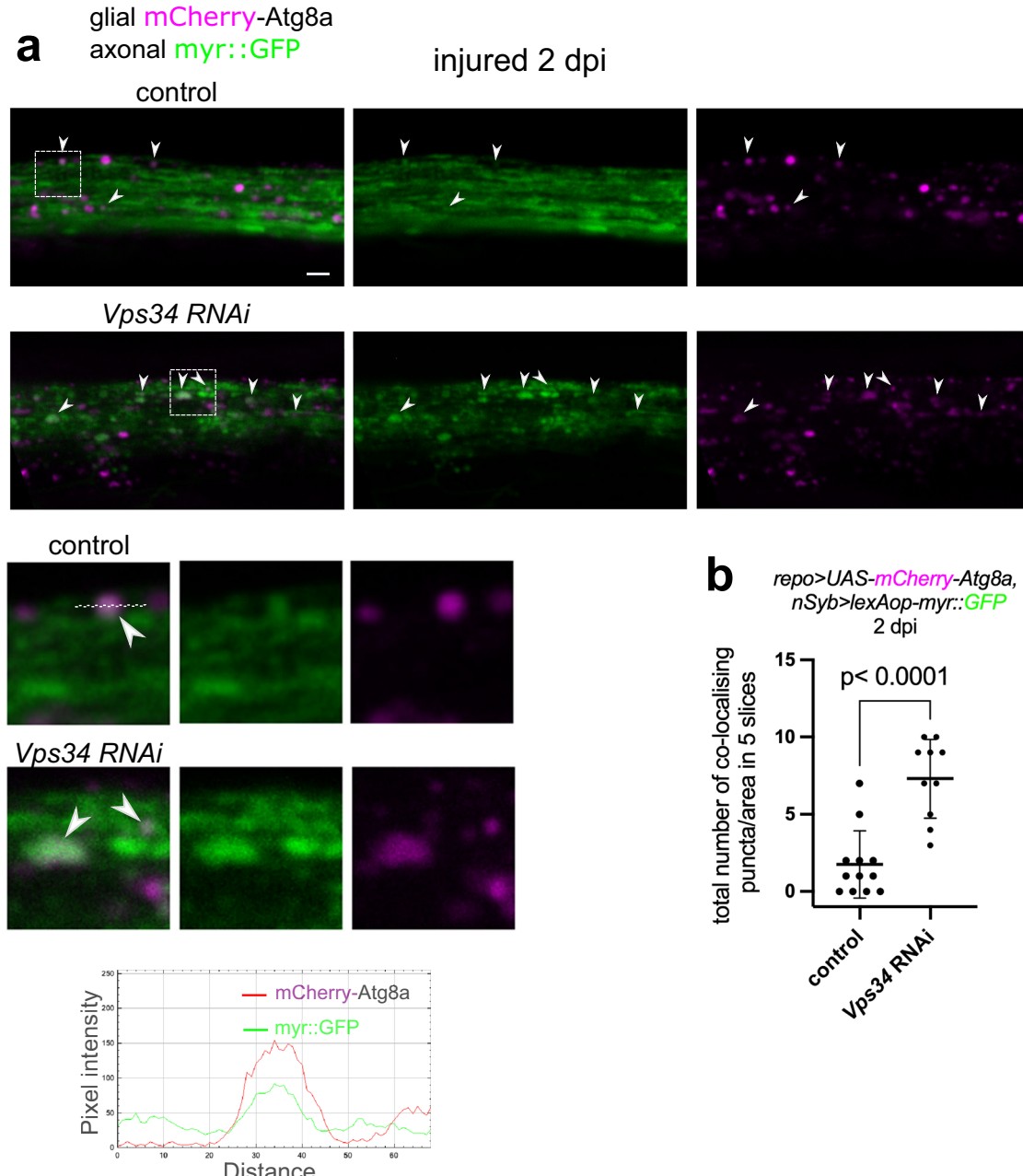

**Fig. 8 | Atg8a is recruited to LAPosomes containing engulfed axon debris, which accumulate upon loss of Vps34. a** Confocal single-slice images of injured wing nerves at 2 days post wing injury (dpi) without or with *Vps34* RNAi in glia with *repo-Gal4*. Glia also express mCherry-Atg8a. Axons are labeled with pan-neuronal myr::GFP (*repo-Gal4> UAS-mCherry-Atg8a, nSyb> lexAop-myr::GFP*). Magnified images of the areas outlined by the rectangles are shown below. Dashed line was used for intensity profile generation. Pixel intensity plots over the indicated distance are shown below for each channel to demonstrate colocalization. Arrowheads point to co-localizing puncta. Scale bar: 2 μm. **b** Quantification of the number of co-localizing mCherry-Atg8a and myr::GFP puncta in five consecutive single-slice images of genotypes in (**a**). Co-localizing puncta were counted in randomly selected 40 × 12 μm areas. Statistical analysis was performed with unpaired, two-tailed Mann–Whitney test. The graph shows the median with 95% confidence intervals. *n* = 12 and 10 biologically independent animals. Source data are provided as a Source Data file.

## Immunostaining of adult brains

Immunostaining of adult brains was performed as described[88] with modifications. A single round of TBI was performed as for lifespan experiments, without repetition. Adult brains were dissected in ice-cold phosphate buffered saline (PBS) and placed immediately in 4% paraformaldehyde in PBS with 0.3% Triton X-100 (PBT) on ice. Brains were fixed for 1 h at RT. After two quick rinses with PBT, brains were washed three times 20 min each. Following blocking in 5% fetal bovine serum (FBS) in PBT for 1 h at RT, brains were incubated with the primary antibody (mouse anti-Drpr 8A1 1:100, Developmental Studies Hybridoma Bank, rabbit anti-GFP 1:1000, Thermo Fisher Scientific A-11122, or rabbit anti-GABARAP/GABARAPL1/GABARAPL2 Abcam ab109364 clone number EPR4805 1:500 for Atg8a) for 3 days at 4 °C in 5% FBS in PBT. Washes were done as before and brains were incubated with the fluorescently labeled secondary antibody (goat anti-rabbit Alexa Fluor 488, A-11034 or goat anti-mouse Alexa Fluor 568, A-11031 all 1:1000 diluted, Thermo Fisher Scientific) in 5% FBS in PBT overnight at 4 °C in darkness. Following washes as above, brains were mounted in Vectashield (Vector Laboratories, H-1000-10).

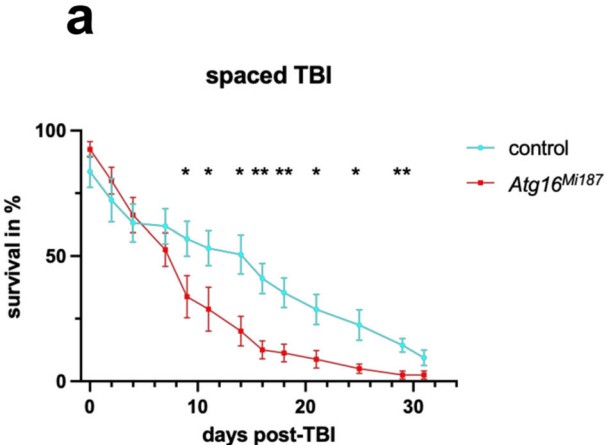

**Fig. 9 | LAP promotes survival after traumatic brain injury. a** Cohorts of 10 flies (1–3 days old) of the indicated genotypes were subjected to traumatic brain injury (TBI) on days 1, 3 and 6, and fly mortality after the last treatment was monitored every 2–3 days. Percentage of surviving flies is shown relative to pre-TBI numbers. $n = 8$, 8 replicates of cohorts. Multiple unpaired two-tailed $t$-tests were performed and the two-stage step-up method of Benjamini, Krieger and Yekutieli was used for correction for multiple comparisons. The corresponding $q$ values from left to right: 0.164120, 0.278307, 0.434948, 0.233553, 0.048106, 0.048106, 0.013568, 0.007739, 0.008695, 0.018284, 0.019754, 0.007739, 0.060341. Asterisk indicates $q < 0.05$. **b** Uninjured controls were treated and analyzed identically to treated flies except for TBI and their survival was assessed in parallel, $n = 12$, 14 replicates of cohorts. Multiple unpaired two-tailed $t$-tests were performed and the two-stage step-up method of Benjamini, Krieger and Yekutieli was used for correction for multiple comparisons. n.s. not significant. The corresponding $q$ values from left to right: 0.995917, 0.701303, 0.667882, 0.667882, 0.667882, 0.667882, 0.667882, 0.667882, 0.667882, 0.880822, 0.995917, 0.880822, 0.667882. **a, b** Data are presented as mean ± standard error of the mean. Source data are provided as a Source Data file.

## Wing injury protocol

Wings were unilaterally injured by complete transection with spring microscissors (Vannas-Tübingen)[9,35,36]. Contralateral wings were left intact and served as a control for nerve integrity in absence of injury. Transection was administered approximately halfway between the tip and the hinge of the wing that left some labeled axons uninjured. Animals were maintained for the indicated time (days post-injury, dpi) after injury and both wings were processed for microscopy.

## Microscopy and image analysis

Wings were mounted pairwise as injured and uninjured from the same animal in Halocarbon oil 27 (Sigma-Aldrich, H8773)[9,35,36]. Samples were imaged immediately after mounting focusing on the proximal wing nerve[9]. Structured illumination fluorescence microscopy was performed on an AxioImager.M2 equipped with an ApoTome.2 structured illumination module (Zeiss) and an ORCA-Flash4.0LT sCMOS camera (Hamamatsu) operated by the ZEN 2.3 software (Zeiss). Illumination was provided by a CoolLED pE-4000 system. For confocal microscopy, we used an LSM800 (Zeiss) inverted laser scanning confocal microscope. The wing nerve was imaged at room temperature using a Zeiss Plan-Apochromat ×63/1.40 NA oil immersion objective with a z-step of 0.25 µm. The same imaging settings were used for all samples on a given microscope. Colocalization experiments (Figs. 4, 7, 8 and S7e, S8, S10) were performed on the LSM800 confocal microscope, all other samples, including brains, were imaged on the AxioImager.M2. Single optical slices of wing nerve images are used for figures.

To quantify axon debris abundance in injured wing nerves, we first verified that debris is progressively cleared from day 2 onwards after injury[9]. Some intact axons are always spared after wing transection and serve as internal control for imaging quality (Fig. S1a). For debris pixel intensity quantification, we used single optical slices. The evaluator was blinded to the identity (genotype, condition) of the image files during quantification. In Fiji (https://fiji.sc, v2.9.0/1.53t)), we selected two 400 × 100 pixel ROI-s covering wholly a section of the fragmented nerve and adjacent, non-GFP+ region, respectively. We measured integrated density of freehand-selected sub-ROI-s inside the 400 × 100 pixel ROI of the nerve, which in total

completely covered any axon debris in this region but minimally contained GFP− background and did not contain uninjured axon fluorescence or autofluorescent cuticle regions. In cases where axon fragments were completely cleared, an ROI of a size similar to previous images was used adjacent to the uninjured axons. We used the exact same selection to measure background fluorescence by measuring integrated density in the second, adjacent non-GFP+ 400 × 100 pixel ROI. To obtain normalized pixel intensity, we subtracted the background integrated density from the nerve debris integrated density.

To quantify GFP-Ref(2)P and 3xmCherry-Atg8a puncta, single-slice images were taken from the same section of the wing nerve where axon debris was quantified. The number of puncta which were clearly distinct from the background and well circumscribed, were counted in Fiji in an ROI of 400 × 200 pixels with the Cell Counter plugin. The evaluator was blinded to the identity (genotype, condition) of the image files during quantification.

To quantify co-localizing puncta, we evaluated the following colocalizations: glial GFP-LAMP1 and CD4::tdTomato+ axon debris, glial Rab5-GFP and CD4::tdTomato+ axon debris, glial Rab7-GFP and CD4::tdTomato+ axon debris, glial Rubicon::mRFP and mCD8::GFP+ axon debris and myr::GFP+ axon debris and glial mCherry-Atg8a colocalization in the wing nerve. We selected five consecutive single confocal slices where co-localizing puncta were counted in a bounding box of 40 × 12 µm in Fiji. Colocalization was scored on well-circumscribed structures where dimensions of GFP-LAMP1+, Rab5-GFP+, Rab7-GFP+, Rubicon::mRFP+ and mCherry-Atg8a+ puncta, respectively, were identical to or slightly larger than the dimensions of the axon fragment that showed full colocalization with them. The sum of colocalization events in five slices is shown. To evaluate Rab7-GFP colocalization with Rubicon::mRFP, a single confocal slice was selected where co-localizing puncta were counted and measured for area and mean pixel intensity in a bounding box of 20 × 12 µm in Fiji. The evaluator was blinded to the identity (genotype, condition) of the image files during quantification. Line plots showing pixel intensity profiles over a distance in color were made with the help of the Fiji plugin RGB Profiler (Christophe Laummonerie, Jerome Mutterer).

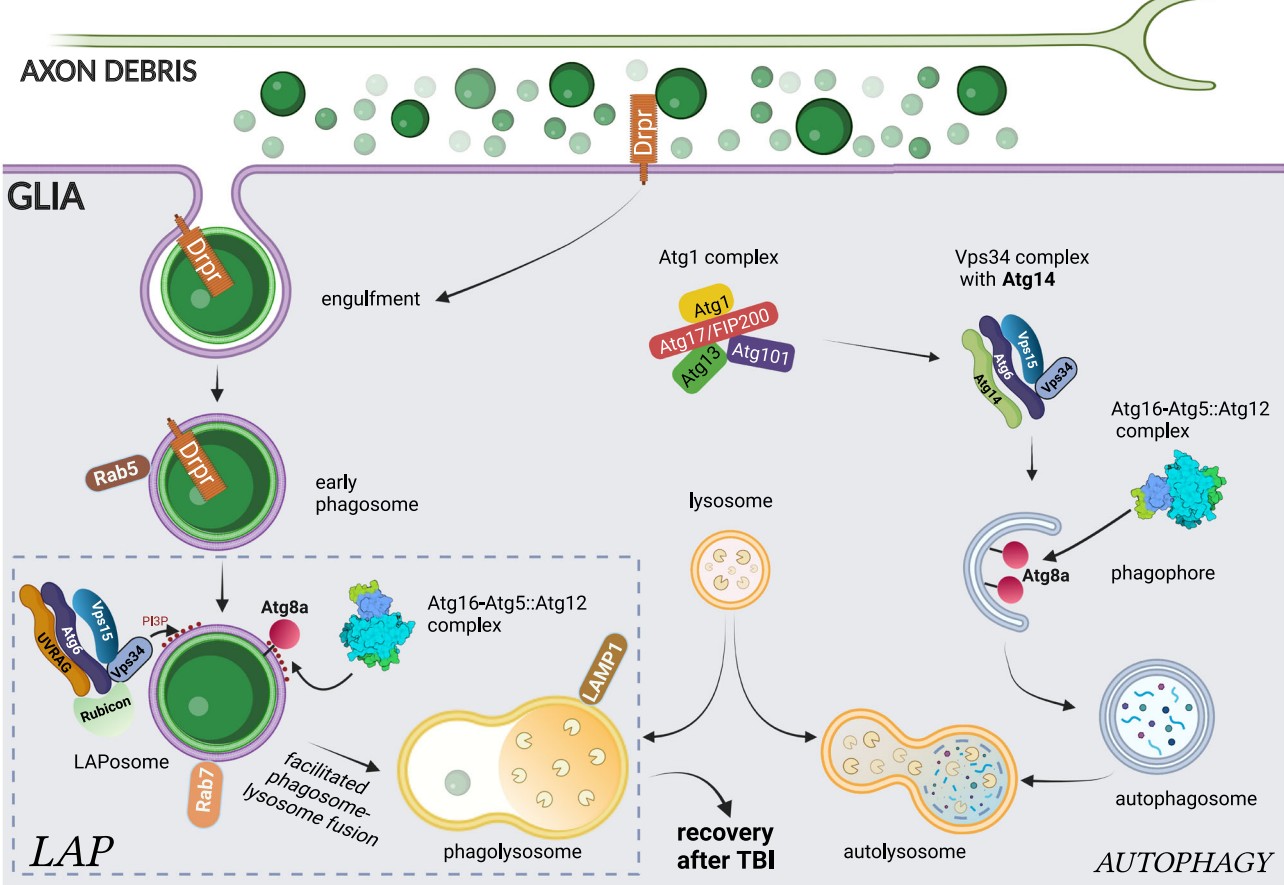

**Fig. 10 | Schematic illustration of the glial LAP mechanism in axon debris clearance.** Fragmenting axons are gradually taken up by glia via Draper-mediated engulfment. Debris is enclosed in a Rab5-positive early phagosome that will mature into a Rab7-positive late phagosome, which may be converted into a LAPosome. LAPosomes are enriched in PI(3)P generated by a Rubicon-associated UVRAG-Vsp34 complex, and lipidated Atg8a is conjugated to their membrane by the Atg5- Atg12-Atg16 complex to facilitate LAPosome maturation and fusion with lysosomes, where axon fragments are degraded. Draper on phagosomes might contribute to activation of LAP in addition to its role in debris recognition and internalization. Autophagy, on the other hand, relies on initiation by the Atg1 complex, followed by PI(3)P generation on phagophores and autophagosomes by an Atg14-containing Vps34 complex, which is also accompanied by Atg8a membrane conjugation.

## Electron microscopy

After rinsing flies in ethanol and distilled water, excess water was removed and animals were transferred to drops of a chemical fixative solution containing 3.2% formaldehyde, 1% glutaraldehyde, 2 mM $CaCl_2$, and 1% sucrose in 0.1 M sodium cacodylate buffer (pH 7.4) on a plastic Petri dish to prevent dispersal of the drop. A small part of the arch region of the L1 wing vein (essentially the boxed region in Fig. 1a) was cut out from the wing in the chemical fixative under a stereo-microscope using a sharp razor blade. L1 wing vein parts were trans-ferred to 1 ml fixative solution and incubated for 2–3 days at 4 °C, and then transferred to 37 °C soft agar (1% in water), which solidified at room temperature. Soft agar blocks containing L1 wing vein parts were then fixed for another day and embedded into Epon resin following the manufacturer's recommendations. Ultrathin sections were cut, post-contrasted and processed for EM analysis following a standard protocol[49].

## Statistical analysis

Experiments were independently repeated twice with similar results and used $n \geq 3$ independent biological replicates except for western blots where $n = 2$. All attempts at replication were successful. Mea-surements were taken from distinct biological samples for individual data points. Error bars and number of data points (biological repli-cates) for each experiment are defined in the figures and figure legends. For debris intensity data, truncated violin plots are shown with median and quartiles containing all data points. To test for normal distribution of data, we used the Shapiro–Wilk normality test ($\alpha = 0.05$). Normally distributed datasets were compared pairwise with unpaired, two-tailed Student's $t$ test and samples in which at least one dataset did not show normal distribution were compared pairwise with unpaired, two-tailed Mann–Whitney test. For more than two datasets, Kruskal–Wallis test for non-normally distributed data and one-way ANOVA for normally distributed data were applied with post hoc tests to correct for multiple comparisons (Dunn's for Kruskal–Wallis and Holm–Šídák's or Šídák's for one-way ANOVA, respectively). In Fig. 8a, multiple unpaired $t$-tests were performed and correction for multiple comparisons was done by the two-stage step-up method of Benjamini, Krieger and Yekutieli. $\alpha = 0.05$ testing level was applied. $p$ values are indicated in the figures. Prism 9.5.1 (GraphPad) was used for statistical analysis and graph generation. No data points were excluded from statistical analysis. Sample size was not predetermined but was similar as in other publications describing axon debris engulfment and autophagy[14,15,38,39,49]. Loss-of-function effects of *drpr* and various autophagy-related genes on different biological processes were taken as a proxy to estimate effect sizes based on mean or median difference of populations. Sample sizes which gave sufficiently large effect sizes in those assays were used in our study.

## Reporting summary

Further information on research design is available in the Nature Portfolio Reporting Summary linked to this article.

## Data availability

All data needed to evaluate the conclusions in this paper are present in the paper, its Supplementary Information and Source Data. Other data associated with the article, such as raw data, are available upon request. The underlying data for Protein Data Bank structure 4NAW of ATG12-ATG5-ATG16N used in Fig. 10 are available at https://www.rcsb.org/structure/4naw. Source data are provided with this paper.

## Materials availability

All materials, *Drosophila* stocks and related information are available from the corresponding authors upon reasonable request.

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

## Acknowledgements

This research was supported by: the Young Researchers' Excellence Programme of the National Research, Development and Innovation Office (NRDIO) (FK132183) (A.S.); the Janos Bolyai Research Scholarship of the Hungarian Academy of Sciences (BO/00078/18) (A.S.); the NRDIO New National Excellence Programme (ÚNKP-20-5, ÚNKP-19-4, ÚNKP-18-4) (A.S.); the Swiss National Science Foundation SNSF Assistant Professor awards (176855 and 211015) (L.J.N.); the International Foundation for Research in Paraplegia (P180) (L.J.N.); SNSF Spark (190919) (L.J.N.); the NRDIO grant KKP129797 (G.J.); the NRDIO grant GINOP-2.3.2-15-2016-00032 (G.J.); and the Biotechnology National Laboratory program of the National Research, Development and Innovation Office (NKFIH-871-3/2020) (G.J.). We thank especially Masaki Oba, Koji Fukui, Kazunori Sango and Mari Suzuki for the *UAS-WT dRubicon::HA* stock and the Rubicon Gateway entry clone, Zsombor Esküdt, Anna Galambos, Róbert Soltész and Dániel Bócsi for their contribution to preliminary experiments, the Bloomington Stock Center, the Vienna *Drosophila* Resource Center, Véronique Monnier, Wanzhong Ge, Hermann Aberle for fly stocks, the Developmental Studies Hybridoma Bank for antibodies and Szilvia Bozsó, Ildikó Kresákné Erdődi and Sarolta Pálfia for technical assistance. We are also grateful to Ildikó Domonkos Racskóné and Petar Lambrev for the access to the Precellys homogenizer and Gábor Csordás, Arindam Bhattacherjee and Tamás Maruzs for fruitful discussions. Figures 2f, 10 and S1 were created with BioRender.com with Fig. 10 using Protein Data Bank structure 4NAW.

## Author contributions

Conceptualization: A.S. and G.J.; Investigation: A.S., V.V., A.S.C., A.H.Ü., S.B., P.B., A.J., K.E.V., and G.J. Resources: A.J. and L.J.N.; Results interpretation: A.S., V.V., A.S.C., A.H.Ü., S.B., K.E.V., P.B., A.J., L.J.N., and G.J.; Writing: A.S. and G.J.; Manuscript revision: A.S., V.V., A.S.C., A.H.Ü., S.B., K.E.V., P.B., A.J., L.J.N., and G.J.; Supervision: A.S. and G.J.; Funding acquisition: A.S., L.J.N., and G.J.

## Funding

## Competing interests

The authors declare no competing interests.
