## [Peer review file · Nature Communications]

REVIEWER COMMENTS

Reviewer #1 (Remarks to the Author):

Szabó et al. conducted an interesting study in which they demonstrated that LC3-associated phagocytosis (LAP) employs a specific subset of autophagy proteins for axon debris clearance after injury of the *Drosophila* wing nerve, an established model to study degeneration of PNS neurons. After wing nerve injury, axons are fragmented, and debris is taken up by glia through Draper (Drpr). The subsequent molecular mechanisms are not well-understood. This manuscript presents a timely and important contribution to our understanding of debris clearance by glia and its intersection with autophagic mechanisms by characterizing a LAP-specific Vps34-dependent mechanism for phagosome maturation.

Specifically, by studying different autophagy mutants, the authors could show that only Atg8a, Atg5 and Atg16, but not the Atg1 complex, are involved in debris removal. They could reproduce this phenotype by silencing these genes in a glia-specific manner. Flies deficient in a LAP-specific domain (Atg16MI) still show debris clearance defects, implying a role for LAP but not canonical autophagy in debris removal. By depleting different Vps34 complex subunits as well as Vps34 they showed that axon debris clearance depends on LAP-specific, but not on the autophagy-specific PI3K complex in glia. Moreover, the Vps34 lipid kinase complex subunit Rubicon is also required for debris elimination. Interestingly, overexpression of Rubicon is sufficient to accelerate debris elimination. Rubicon as well as Atg8a are recruited to Rab7-positive debris-containing phagosomes.

Taken together Szabó et al. provide new insights in axon debris clearance in PNS neurons based on a variety of genetic approaches. Each of their experiments address different parts of the pathway, building a model step by step. The experiments are conducted carefully and the manuscript overall represents a solid advance that warrants publication with high priority. We have only minor quibbles that should be addressed, largely to make it easier for the reader to follow experimental rationales.

Minor:

- A schematic of the pathway would be useful to understand where each experiment fits in.

-

- Manuscript page 9, 10; Figure 4: In this critical experiment the authors aim to analyze glial phagosome-lysosome fusion by looking at the colocalization of a glia-expressed LAMP1 and a neuronally expressed tdtomato. However, in our reading of the results the manipulation of atg16 knock-down in glia causes a defect in uptake prior to a potential phagosome-lysosome fusion event inside the glial cells – so it is not really surprising to find less colocalization. In order to ensure that only tdtomato that was taken up in glia is included in this quantification, it would be great to label the glia independently of the LAMP1 marker. Can the author quantify how much of the tdtomato is engulfed and taken up at all in this experiment?

-

- Fig. 1 b: 5 dpi, Atg8a channel: there is text (10 dpi) in this image
- Fig. 1 b and c; Fig. S1 a: show all genotypes for all conditions (Atg8a for 2 dpi and 10 dpi, Atg8 rescue for 2 dpi and 5 dpi)
- Why was it necessary to use trans-heterozygous Atg16 mutants (Fig. 1), an Atg17/FIP200 hypomorph allelic combination (Fig. 2) and a drpr heterozygous null mutant? This should be discussed in the manuscript or in the materials and method section.
- Page 8; line 20-21; Fig. 3: “Similarly, knockdown of the autophagosomal Qa SNARE Syntaxin 17 (Syx17)⁴⁰ did not impair debris elimination” – the relevance of this result should be discussed in the manuscript.
- *Drosophila* should be written in italic. (page 5, line 12; page 18, line 20; page 19, line 1)
- Page 6, line 4: reference 9 is in brackets and not as the other references above the text.

Reviewer #2 (Remarks to the Author):

Szabo et al examine the role of LC3-associated phagocytosis (LAP) in the degradation of axonal debris by glial cell in the fly wing. They examine the well-established pathway targeting of the mediators of the LAP pathway. They established (using fly orthologues) that LAP depends on the LC3, the Atg5 complex, UVRAG and Rubicon; concluding that LAP is involved in recovery of injured neuron by clearing debris.

The studies are well performed, the data are clearly described, and the conclusions are logical. Although the data readout is pretty much the same in all figure. It would be helpful to visualize the glial cells are higher magnification and even examine the putative phagosomes at the TEM level to determine in the glial cells actually contain single membrane phagosomes as opposed to double membrane autophagosomes.

Reviewer #3 (Remarks to the Author):

In this manuscript, Szabó and colleagues investigated the role of autophagy and related pathways in axon debris clearance after injury in the *Drosophila* wing nerve, which is an established model for Wallerian degeneration in the peripheral nervous system (PNS). CRISPR-Cas mediated mutants deficient

for the Atg8 α lipidation system as well as Atg8a Δ 4 mutant flies showed defects in debris clearance compared to wild-type flies. These defects were rescued by an Atg8a::GFP tagged transgene. Then the authors showed that mutants for the Atg101 and Atg17 genes encoding components of the Atg1 complex did not show debris accumulation. In contrast, a specific domain with WD repeats, of Atg16 which functions in LC3-associated phagocytosis (LAP) appears to play a role in debris clearance. Furthermore, heterozygotes mutants for the Draper receptor (Drpr) showed debris accumulation, similar to Atg8a Δ 4 mutants. However, double mutants for Atg8a and Drpr did not exhibit any additive defect, suggesting that these genes most likely function in the same pathway. Further analysis showed that a subset of autophagy genes, including the autophagosomal Qa SNARE Syntaxin 17 (Syx17), which are known to be required for autophagy in glia are dispensable for debris clearance in the axons. On the other hand, the Vps34 complex II is crucial for debris removal since knockdown of its components such as Rubicon and UVRAG led to debris accumulation in glia. Consistently, Rubicon overexpression was sufficient to mediate debris endocytosis. Finally, the authors showed that Rubicon is recruited to the phagosome where it colocalises with RAB7 under both normal conditions and upon injury. This finding combined with Atg8a recruitment to the phagophore in injured axons suggest that degradation of axon debris in glia is mediated by LAP.

Comments

A key point of the study is that Rubicon binds to RAB7. However, this finding is hardly new (Bhargava et al, 2020, *Biophysics and Computational biology*, 117 (29) 17003-17010). Nevertheless, since the crystal structure of the Rubicon RH domain in complex with Rab7-GTP was recently resolved, it would be interesting to test whether *Drosophila* mutants lacking this RH domain show defective debris clearance.

The study suggests a crucial role for Rubicon in mediating LAP activation in the injured nervous system, thereby facilitating removal of debris during necrosis. This statement should be tested experimentally in *Drosophila* models of neurodegeneration. A previous study has already shown that LAP is compromised in primary astrocytes of Huntington's disease mouse models (Wakida et al. 2022, *Journal of Huntington's Disease* 11, 25–33). In other words, the authors could also test whether enhanced Rubicon function in glia can facilitate the removal of dead brain cells in *Drosophila* models of acute and chronic neurodegeneration by stimulating LAP activity, as mentioned in the last sentence of Discussion.

Although the model for Wallerian degeneration in the PNS used here is well-established, monitoring the responsiveness of about 40 nerve axons in the wings (glutamatergic neurons) narrows the window of analysis. To further justify their findings and carefully assess the role of ATG8 in the degenerative progress, additional fluorescent markers such as mCD8-CherryRFP should be used.

The authors should use additional biochemical assays to verify the contribution of either autophagy or LAP on debris elimination instead of overly relying on confocal imaging. For instance, in Figure 1B and 1D

protein levels of Atg8, Atg16 and Atg5 after 2, 5 and 10 days post wing injury should be examined. It would also be interesting to show whether and when Atg8-positive puncta are formed upon injury.

In Figure 2A the authors show that mutated Atg8aG116* which fails to undergo any lipidation event also leads to debris accumulation. It would be interesting to show with imaging whether at the site of injury Atg8aG116* flies exhibit a diffuse cytosolic expression.

In Figure 2A and 2D the authors claim that the ATG1 complex is dispensable for axon debris removal although it is required for autophagic process. This statement is not supported by the findings provided. Firstly, the authors should show whether Atg101 and Atg7 are present in both injured and uninjured wing L1 veins. Next, it will be interesting to check whether loss of Atg101 or Atg17 disfavors the presence of Atg8-positive puncta at the site of injury

Figure 2D and 3E: The authors show that blockade of autophagy components leads to reduction of Atg8a positive puncta and an increase in Ref(2)P puncta in uninjured neurons. It would be interesting to show what happens on Atg8a and Ref(2)P puncta in control flies upon injury. Do they respond? As autophagy is not induced and LAP takes over, Atg8A should increase and Ref(2)P should stay static. Is this the case?

In Figure 2H the authors state that LAP and canonical autophagy act through the same pathway for debris engulfment. To make a definitive statement that couples the LAP with the canonical autophagy, more evidence has to be provided. In my view, a time course analysis of debris elimination at days 2, 5 and 10 dpi should be included upon single and double inhibition of drpr and Atg8. Moreover, double mutants of drpr with Atg101 and Atg17, and subsequent examination of remaining axon debris will give additional information regarding the mode of interaction between LAP and autophagy.

Figure 2H,I: the results show that the combination of drpr Δ 5 and Atg8a Δ 4 actually reverse the effects of each single treatment (Although the quantification analysis shows a statistical difference between . This could indicate that they act redundantly in the clearance of axonal debris. Is LAP activated when Drpr is mutated and vice versa? What is the sequence of events? Do they happen simultaneously?

In the experimental design in Fig.4 authors should have used the Atg16MI mutants and not Atg16 RNAi that would impact both autophagy and LAP. In the same Figure, authors don't show whether the depicted debris is indeed engulfed by phagosomes.

There is no direct evidence demonstrating that phagosomes contain axonal debris. Authors should use electron microscopy or specialized fluorescence microscopy methods to convincingly show this. Electron

microscopy would be optimal as, through it, the authors could additionally show that debris are contained by a single membrane formation-as expected for the phagosome- and not in double membrane formations (that would be autophagosomes), further strengthening their initial observations that autophagy is dispensable.

The claim that 2 days post injury a different type of Rubicon-Rab7 double positive vesicle appears only taking into account the size is unsubstantiated. The same stands for the claim that this differentiated type of vesicle contains less Rubicon due to limited Rubicon association. This could just be due to decreased protein expression in the injured compared to the uninjured. To claim that a different type of vesicle is formed authors should have isolated vesicles from the two time points and perform proteomic analysis to check their protein constituents or/additionally they could have manually checked the presence or absence of endosomal markers in these (larger) structures.

Fig. 6A: The dotted expression pattern in this figure is not sufficient to support that Rubicon is localized to vesicles. What kind of vesicles? No vesicular marker is used to support this statement. Colocalization of glial Rubicon::mRFP1 vesicles and GFP-labelled axon debris does not seem prominently affected, when comparing wing nerve injury and random Rubicon puncta colocalization on GFP-labelled axons without injury. Puncta colocalization quantification is missing. Zoomed areas could be accompanied by single channel images in order to closely assess and evaluate the morphology of colocalized puncta.

Another issue relevant to Figure 6A and 6B pertains to the effect of injury in the total protein levels of Rubicon in the glia. While in Figure 6A Rubicon is increased in injured versus uninjured the opposite is shown in Figure 6B.

The authors should show mechanistically whether Rubicon blocks endocytosis and if yes, then how are the rest of the results (regarding the effect of Rubicon on debris elimination) explained? What is the effect of Rubicon on LAP in glial cells?

Minor

Most experiments performed do not include the control uninjured axons, it will be interesting and more informative to be included at least in the main experiments. The authors should include both positive and negative controls at least for the most essential experiments performed.

Figure 1B is not very informative. It should be combined with a bright field image so that the wing morphology becomes more recognizable.

In Figure 3A, contrary to what is mentioned in the text, atg1 inhibition seems to cause debris removal defects.

In Figure 6A co-localization is not obvious. Author should perform statistical analysis

Figure 2E: It is not clear why the authors choose to use two different statistical tests for the comparison of Atg101 and Atg17/FIP200 mutants as opposed to Atg5 mutants with the control .

Figure 3a: Axonal debris are still visible in atg1 RNAi flies, with an incidence higher than in control flies. This phenomenon is not recapitulated in the quantification panel 3b. Maybe a different image should be used.

Figure 7: The image of axonal debris in this strain is completely different than that in the mCD8::GFP strains. There are no obvious puncta or uninjured neurons. How do the authors explain this difference?

Reference 85 is missing.

The introduction and discussion are rather long and could be more focused.

Reviewer #4 (Remarks to the Author):

In this manuscript, Szabó et al claim that glial LC3-associated phagocytosis (LAP) promotes clearance of degenerating axons. The *Drosophila* LC3 homologue is Atg8. They claim that LAP depends on Atg8 EP lipid conjugation, the Atg8 conjugation system formed of Atg5, Atg12 and Atg16, and on the Vps34 lipid kinase subunits UVRAG and Rubicon, but not Atg14 or the Atg1 kinase complex members Atg101 or Atg17. They show that knock-down in glia of UVRAG, rubicon or Vps34 all prevent debris clearance. They claim that Atg8 and Drpr work in the same pathway; that Atg8 and Rubicon are recruited to Rab7+ phagosomes; that Rubicon normally inhibits endocytosis, which is similar to phagocytosis, in un-injured samples; that knock-down of Rubicon in glia prevents debris clearance; that a decrease in Rubicon levels accompanies the maturation of LAPosomes; whereas over-expression of Rubicon in glia accelerates axon debris clearance.

The following points need resolving and suggestions are made for improvement:

1. Evidence unclear on the involvement of Atg8 in multiple aspects of study:

If the wing is cut and thrown away, as in Figure 1, how is it possible that 'some axons are spared by the injury' as stated in the methods section? Is it that some axons do not undergo Wallerian degeneration, and is this affected differentially by the different genotypes? The quantifications described in the methods say that a free-hand ROI excluding the axons is chosen to quantify axonal debris. However, remaining axons differ across genotypes, eg there seem to be more remaining axons in both Atg8 LOF and GOF in Figure 1. For example, remaining axons in Fig.1B vs control could suggest that Atg8 is required for Wallerian degeneration in axons. Quantifying the remaining axons could reveal whether Atg8 is required for Wallerian degeneration or not.

Based on Figure 2I the authors conclude that Atg8 and Drpr function in the same pathway, because the double LOF *atg8 drpr*⁻ gives a not-statistically significant difference vs Atg8 LOF. However: 1, the double is statistically significantly different from *drpr*^{-/+} alone; 2, the sample size differs in Atg8D4 (n=16) vs the double (n=12); 3, the median in the double is virtually the same as for the control and differs greatly from the medians for both *drpr*⁻ and Atg8D4 mutants. Thus, the graph evidence indicates that the sample size for the double must be increased to at least n=16 as it could become statistically significantly different from Atg8D4 alone. This is important, as it could indicate that in the double *drpr*^{-/+} Atg8D4 there is a rescue phenotype. Together with the above, it could indicate that Atg8 is required in axons for Wallerian degeneration.

The authors then test the involvement of the conjugation system Atg5 and Atg16 and of Atg1, Atg13 and Atg 17 by knocking them down in glia, and they conclude that the conjugation system is required but not Atg1 complex. However, they did not test Atg8. It is essential that they test *repoGAL4>UAS-Atg8-RNAi*.

2. Evidence must be provided that RNAis and glial-targeted CRISPR/Cas9 are being knocked-down, with qRT-PCR. Instead, they use an autophagy assay as evidence of RNAi knock-down in experiments which address the question of whether autophagy is involved or not. This is indirect, circular and not appropriate, and the evidence of RNAi knock-down must be direct and specific to the targeted gene.

3. The function of rubicon is unclear: if Rubicon normally inhibits endocytosis, and Rubicon signal intensity and puncta decrease after injury reflecting the maturation of LAPOsomes, then why does rubicon RNAi prevent cell debris clearance, and over-expression increases it?

4. Genetic background contribution to phenotypes must be ruled out by using trans-heterozygous combinations of 2 different alleles: Fig.1,2,3,5.

5. Statistical analysis for figures that contain graphs with more than 2 genotypes (e.g. Fig.1I, Fig.2B etc.): it would appear the p values on the figures correspond to the multiple comparisons correction test results? This information must be provided in the figure legends. They should include in the figure legend the p value for the Kruskal Wallis ANOVA and/or One Way ANOVA, the multiple comparisons corrections tests used and then indicate what test the p values on the graphs refer to.

6. Writing could improve. From the background information in the introduction, the novelty of this study is not always clear. Too much background information is also provided within results, in places.

7. An illustration figure of the pathways under study would really help.

Applies to most figures:

1. Non-injured controls must be provided in figures, as evidence that injury took place.

2. Would be best to avoid writing the genotype over image data as they obscure the data in multiple cases, specially with long genotypes.

3. Images are given as Z-projections, and quantifications are mostly given of single sections. Thus, images are not representative of the data given in the graphs. Instead, quantifications must reflect the image data provided, and vice-versa. If the quantifications are on single sections, then the images ought to be too.

4. Figures must contain complete genotypes, currently some (eg Figure 2D,E) do not.

5. Figures claiming co-localisation must include split channels of their higher magnification views, otherwise evidence is not visible: Figure 4, 6, 7.

Response to Reviewers

We would like to thank the expert Reviewers for their suggestions on how to improve our manuscript. Accordingly, we have addressed all their comments and included further explanations and new data as appropriate. Please see below for details.

Reviewer #1 (Remarks to the Author):

Szabó et al. conducted an interesting study in which they demonstrated that LC3-associated phagocytosis (LAP) employs a specific subset of autophagy proteins for axon debris clearance after injury of the *Drosophila* wing nerve, an established model to study degeneration of PNS neurons. After wing nerve injury, axons are fragmented, and debris is taken up by glia through Draper (Drpr). The subsequent molecular mechanisms are not well-understood. This manuscript presents a timely and important contribution to our understanding of debris clearance by glia and its intersection with autophagic mechanisms by characterizing a LAP-specific Vps34-dependent mechanism for phagosome maturation.

Specifically, by studying different autophagy mutants, the authors could show that only Atg8a, Atg5 and Atg16, but not the Atg1 complex, are involved in debris removal. They could reproduce this phenotype by silencing these genes in a glia-specific manner. Flies deficient in a LAP-specific domain (Atg16MI) still show debris clearance defects, implying a role for LAP but not canonical autophagy in debris removal. By depleting different Vps34 complex subunits as well as Vps34 they showed that axon debris clearance depends on LAP-specific, but not on the autophagy-specific PI3K complex in glia. Moreover, the Vps34 lipid kinase complex subunit Rubicon is also required for debris elimination. Interestingly, overexpression of Rubicon is sufficient to accelerate debris elimination. Rubicon as well as Atg8a are recruited to Rab7-positive debris-containing phagosomes.

Taken together Szabó et al. provide new insights in axon debris clearance in PNS neurons based on a variety of genetic approaches. Each of their experiments address different parts of the pathway, building a model step by step. The experiments are conducted carefully and the manuscript overall represents a solid advance that warrants publication with high priority. We have only minor quibbles that should be addressed, largely to make it easier for the reader to follow experimental rationales.

Thank you for the positive evaluation of our work.

Minor:

- A schematic of the pathway would be useful to understand where each experiment fits in.

Thank you for this suggestion, we have added the schematic of the pathway in Figure 10.

- Manuscript page 9, 10; Figure 4: In this critical experiment the authors aim to analyze glial phagosome-lysosome fusion by looking at the colocalization of a glia-expressed LAMP1 and a neuronally expressed tdtomato. However, in our reading of the results the manipulation of atg16 knock-down in glia causes a defect in uptake prior to a potential phagosome-lysosome fusion event inside the glial cells – so it is not really surprising to find less colocalization. In order to

ensure that only tdtomato that was taken up in glia is included in this quantification, it would be great to label the glia independently of the LAMP1 marker. Can the author quantify how much of the tdtomato is engulfed and taken up at all in this experiment?

Thank you for raising this point. We agree that these control experiments are important to better support a specific role for Atg16 in phagosome-lysosome fusion. We have thus GFP-labelled the glial plasma membrane, or glial Rab5+ early phagosomes, or Rab7+ late phagosomes, respectively, and concurrently axon membranes with tdTomato. Upon injury, we see that tdTomato+ axon debris is contained in phagosomes forming from the glial plasma membrane (Fig. S7e). We found that tdTomato+ axon debris reaches glial Rab5+ and Rab7+ vesicles (that is, phagosomes/LAPosomes), and there is no difference in colocalization rates between control and glial Atg16 knockdown wing nerves (Fig. S8), so we don't think that engulfment is impaired. This is now also supported by our ultrastructural analysis (please see new Fig. 5).

- Fig. 1 b: 5 dpi, Atg8a channel: there is text (10 dpi) in this image

We thank the reviewer for noticing this, we have removed the extra text from this panel.

- Fig. 1 b and c; Fig. S1 a: show all genotypes for all conditions (Atg8a for 2 dpi and 10 dpi, Atg8 rescue for 2 dpi and 5 dpi)

Thank you for this suggestion, we have added the requested new data to the manuscript. Atg8a^{Δ4} was already shown at both 2 dpi and 10 dpi in the first manuscript version, so we assume that Reviewer 1 meant the Atg16 mutant combination, which we also show now at 2 and 10 dpi, along with Atg8a rescue in revised Fig. 1b, c.

- Why was it necessary to use trans-heterozygous Atg16 mutants (Fig. 1), an Atg17/FIP200 hypomorph allelic combination (Fig. 2) and a drpr heterozygous null mutant? This should be discussed in the manuscript or in the materials and method section.

Thank you for suggesting to clarify these. We have added the following explanations to the text:

“We first studied axon fragmentation and persistence in the viable Atg8a^{Δ4} deletion mutant and a trans-heterozygous Atg16 mutant combination (Atg16^{Δ67/Δ129}) that we chose to rule out genetic background effects on the potential phenotype.”

“We thus selected two viable mutants for genes encoding Atg1 complex members: an Atg101 null (Atg101^{Δ6h}) and an Atg17/FIP200 severe hypomorph allelic combination (Atg17^{Δ130}/Atg17^{MiMIC}, note that Atg17 null mutants are pupal lethal so unsuitable for wing injury analysis) and gauged whether debris clearance depends on Atg1 kinase complex activity.”

“In Draper homozygous null mutant flies, glia fail to even make contact with axon debris^{8,10}. Strikingly, heterozygosity for drpr is enough to strongly interfere with axon debris phagocytosis, so we decided to use this sensitized genetic background to test if halving the level of Draper has any further effect on axon debris clearance in LAP mutants.”

- Page 8; line 20-21; Fig. 3: “Similarly, knockdown of the autophagosomal Qa SNARE Syntaxin 17 (Syx17)⁴⁰ did not impair debris elimination” – the relevance of this result should be discussed in the manuscript.

Thank for the suggestion to better explain this result. We added this text: “Since knockdown of the autophagosomal Qa SNARE Syntaxin 17 (Syx17) did not impair debris elimination (Fig. 3k,l, S5e), which severely impairs autophagosome-lysosome fusion in Drosophila tissues, this also supports our model that canonical autophagy acting through Syx17 is dispensable for debris clearance.”

- Drosophila should be written in italic. (page 5, line 12; page 18, line 20; page 19, line 1)

Thank you, we corrected these.

- Page 6, line 4: reference 9 is in brackets and not as the other references above the text.

Thank you for the remark, we corrected this.

Reviewer #2 (Remarks to the Author):

Szabo et al examine the role of LC3-associated phagocytosis (LAP) in the degradation of axonal debris by glial cell in the fly wing. They examine the well-established pathway targeting of the mediators of the LAP pathway. They established (using fly orthologues) that LAP depends on the LC3, the Atg5 complex, UVRAG and Rubicon; concluding that LAP is involved in recovery of injured neuron by clearing debris.

The studies are well performed, the data are clearly described, and the conclusions are logical. Although the data readout is pretty much the same in all figure. It would be helpful to visualize the glial cells are higher magnification and even examine the putative phagosomes at the TEM level to determine in the glial cells actually contain single membrane phagosomes as opposed to double membrane autophagosomes.

*Thank you for the positive overall evaluation of our manuscript. We agree that ultrastructural analysis remains a very important part of vesicle trafficking research. Recent publications already established the ultrastructure of uninjured wing nerves, which was possible to do by using a special equipment (Pelco BioWave Pro, Ted Pella) to penetrate the insect cuticle during sample processing (e.g., Chung Neuron, 2020 May 20;106(4):589-606.e6). With no access to such equipment, we had to optimize sample preparation and processing for EM analysis, which we eventually solved by cutting out a tiny piece of the L1 wing vein during sample preparation. The ultrastructure of the uninjured wing nerve using this protocol is the same as in previous papers, and we also investigated control and glial Atg5 knockdown wing nerves 2 days after injury (please see Figure 5 in the revised manuscript). We detected many axon fragments outside of glia as well as in early and late phagosomes within glia in both genotypes. As expected based on our colocalization and genetic analyses, phagocytosed axon fragments are surrounded by the axon membrane and the single phagosome membrane in both genotypes. As explained in **Fig. 1 for Reviewers**, engulfed axon fragments would be surrounded by 4 membranes in glial cells if they were subsequently captured into autophagosomes: 1. the neuron/axon plasma membrane of*

the axon fragment (axolemma), 2. the glial plasma/phagosome membrane, and 3-4. the two membranes of the autophagosome. We never saw any 4-membrane vesicles in glia, further supporting the role of LAP in axon fragment clearance. Interestingly, axon fragment breakdown in the late phagosomes of *Atg5* knockdown glia appear less condensed than those of injured controls, further supporting the endosome maturation/lysosomal fusion defect.

Figure 1 for Reviewers. Schematic representations of: a. LAP: phagocytosis of axon fragments (surrounded by the axon membrane/axolemma and the glial plasma membrane-derived phagosome membrane) and subsequent conjugation of *Atg8a* to the phagosome membrane, versus b. phagocytosed axon fragments are captured into double-membrane autophagosomes, producing vesicles with 4 surrounding membranes.

Reviewer #3 (Remarks to the Author):

In this manuscript, Szabó and colleagues investigated the role of autophagy and related pathways in axon debris clearance after injury in the *Drosophila* wing nerve, which is an established model for Wallerian degeneration in the peripheral nervous system (PNS). CRISPR-Cas mediated mutants deficient for the *Atg8a* lipidation system as well as *Atg8a* Δ 4 mutant flies showed defects in debris clearance compared to wild-type flies. These defects were rescued by an *Atg8a::GFP* tagged transgene. Then the authors showed that mutants for the *Atg101* and *Atg17* genes encoding components of the *Atg1* complex did not show debris accumulation. In contrast, a specific domain with WD repeats, of *Atg16* which functions in LC3-associated phagocytosis (LAP) appears to play a role in debris clearance. Furthermore, heterozygotes mutants for the Draper receptor (*Drpr*) showed debris accumulation, similar to *Atg8a* Δ 4 mutants. However,

double mutants for Atg8a and Drpr did not exhibit any additive defect, suggesting that these genes most likely function in the same pathway. Further analysis showed that a subset of autophagy genes, including the autophagosomal Qa SNARE Syntaxin 17 (Syx17), which are known to be required for autophagy in glia are dispensable for debris clearance in the axons. On the other hand, the Vps34 complex II is crucial for debris removal since knockdown of its components such as Rubicon and UVRAG led to debris accumulation in glia. Consistently, Rubicon overexpression was sufficient to mediate debris endocytosis. Finally, the authors showed that Rubicon is recruited to the phagosome where it colocalises with RAB7 under both normal conditions and upon injury. This finding combined with Atg8a recruitment to the phagophore in injured axons suggest that degradation of axon debris in glia is mediated by LAP.

Comments

A key point of the study is that Rubicon binds to RAB7. However, this finding is hardly new (Bhargava et al, 2020, Biophysics and Computational biology, 117 (29) 17003-17010). Nevertheless, since the crystal structure of the Rubicon RH domain in complex with Rab7-GTP was recently resolved, it would be interesting to test whether Drosophila mutants lacking this RH domain show defective debris clearance.

Thank you for this note. We think that the most important messages of our work are 1. the in vivo role of LAP in the glial clearance of injured axon fragments in the nervous system of a complete animal, 2. genetic delineation of the LAP pathway in this setting, 3. precise establishment of the clearance defect in LAP mutant glia, and 4. now our new data showing that LAP-defective animals cannot properly survive traumatic brain injury (please see Figure 9). We find the reviewer's suggestion interesting and probably Rab7 binding is important for Rubicon's function on late endosomes/late phagosomes, we feel that the investigation of debris clearance in RH domain-deficient Rubicon mutants is beyond the scope of our in vivo study because our work is not focusing on the interaction between Rab7 and Rubicon. Rubicon modulation is just one of the many genetic tools that we use to study LAP in our in vivo model, and generating a new knockin animal would be a project on its own. To acknowledge this interaction, we added this text to the revised discussion: "Rubicon's C-terminal RH domain directly binds to Rab7 (Bhargava 2020 PNAS), which explains its presence on Rab7 vesicles."

The study suggests a crucial role for Rubicon in mediating LAP activation in the injured nervous system, thereby facilitating removal of debris during necrosis. This statement should be tested experimentally in Drosophila models of neurodegeneration. A previous study has already shown that LAP is compromised in primary astrocytes of Huntington's disease mouse models (Wakida et al. 2022, Journal of Huntington's Disease 11, 25–33). In other words, the authors could also test whether enhanced Rubicon function in glia can facilitate the removal of dead brain cells in Drosophila models of acute and chronic neurodegeneration by stimulating LAP activity, as mentioned in the last sentence of Discussion.

Thank you for this note. We think that using Rubicon gain- or loss-of-function in a disease setting is problematic because Rubicon inhibits both autophagy and endocytic degradation (which was established in 2009 and confirmed in several papers since then), while it promotes LAP. The exact details of this molecular regulation are still open questions. We would like to emphasize that Rubicon knockout animals are long-lived because basal autophagy is elevated in the absence

of this negative regulator (Nakamura Nat. Commun. 2019 Feb 19;10(1):847), so the effects of its modulation are not specific for LAP. The autophagy block upon Rubicon overexpression complicates the interpretation of neurodegeneration experiments where symptoms and cell death are occurring on a longer timescale (several weeks or a month in adult flies).

To demonstrate the in vivo significance of glial LAP in the CNS, we used a repetitive mild traumatic brain injury (TBI) protocol that leads to acute and chronic neurodegeneration (Barekat Sci Rep., 2016 May 4;6:25252). We now show that TBI induces axon fragmentation (Fig. S3a), punctate Draper localization, a marker of glial phagosomes (Ziegenfuss Nat Neurosci. 2012 Jun 17;15(7):979-87.) (Fig S3b), and Atg8a lipidation (Fig. 2h) in the brain. We then analyzed the physiological importance of LAP in a situation when canonical autophagy remains unaffected using the WD40 domain-specific mutant of Atg16, which specifically disrupts LAP, but it does not affect autophagy (Nagy Development. 2017 Nov 1;144(21):3990-4001. and this study, see also Fletcher EMBO J. 2018 Feb 15;37(4):e97840. for mammalian data). We exposed control and mutant animals to TBI on days 1, 3 and 6, which caused much higher rates of mortality in mutants compared to identically treated control flies (Figure 9). Importantly, Atg16 WD40 mutants that are not subjected to TBI had similar mortality as control flies during the same period (Figure 9). These data suggest that LAP is important for recovery after brain injury to ensure survival.

Although the model for Wallerian degeneration in the PNS used here is well-established, monitoring the responsiveness of about 40 nerve axons in the wings (glutamatergic neurons) narrows the window of analysis. To further justify their findings and carefully assess the role of ATG8 in the degenerative progress, additional fluorescent markers such as mCD8-CherryRFP should be used.

Thank you for these suggestions. Please note that following the disappearance of acid-sensitive GFP fluorescence is a standard assay in the autophagy field, because red fluorescent proteins remain fluorescent in lysosomes. The different acid sensitivity of GFP vs Cherry type proteins serves as the basis of the commonly used “flux” reporters, such as mCherry-GFP-Atg8a/LC3 for autophagy. Similar reporters are used to track endocytic degradation in the fly CNS (Jin Curr Biol, 2018 Apr 2;28(7):1027-1038.e4). Thus, axonal Cherry/RFP is not suitable for the study of lysosomal clearance.

We chose to follow glutamatergic axon fragments because this is standard in the wing nerve injury field based on papers from the laboratories of Marc Freeman, Yanshan Fang, and one of the authors of this manuscript, Lukas Neukomm (Neukomm PNAS, 2014 Jul 8;111(27):9965-70., Neukomm Neuron, 2017 Jul 5;95(1):78-91.e5., Wang Sci Adv, 2019 Feb 13;5(2):eaav4971., Farley PNAS, 2018 Feb 6;115(6):1358-1363, Rosell 2022, eLife, 2022 Dec 23;11:e80245). Marking only a subset of neurons is usually necessary to enable proper counting of fragments: please note that 5 times more axons/fragments are labeled in the same volume when using a pan-neuron driver (e.g., nSyb-Gal4). Nevertheless, we do have loss-of-function data when we follow all neurons in our manuscript: please see Fig. 4 (Atg16 knockdown in glia disrupts LAMP1 colocalization with axon debris), Fig. 8 (Vps34 knockdown in glia prevents the clearance of glial Atg8a+ phagocytosed axon fragments), and the new ultrastructural data in Figure 5.

The authors should use additional biochemical assays to verify the contribution of either autophagy or LAP on debris elimination instead of overly relying on confocal imaging. For instance, in Figure 1B and 1D protein levels of Atg8, Atg16 and Atg5 after 2, 5 and 10 days post wing injury should be examined. It would also be interesting to show whether and when Atg8-positive puncta are formed upon injury.

Thank you for this note. Actually, confocal microscopy was the only assay used in previous papers of the wing injury field (please see examples in our response to your previous comment). Nonetheless, we carried out western blots as suggested. Biochemical evaluation of Atg protein levels in severed wings was not feasible because it provides very scarce material. To address this comment, we used the repetitive TBI model again to inflict injury to flies and collected heads at 2-, 5-, and 10-days post injury. Western blots of head extracts showed no change in the levels of Atg5, the Atg5-12 conjugate, and Atg16 isoforms upon injury (Fig. S3d, e). As expected, the lipidated (membrane-conjugated) form of Atg8a did show elevated levels after injury (Fig. 2h), concomitant with LAP activation. Additionally, we see an increase of mCherry-Atg8a puncta in glia after injury, supporting LAP initiation (Fig. S3f), especially when it is taken into account that the selective autophagy cargo p62/Ref(2)P in glia does not show a significant change in response to injury (Fig. S3g). Please note that we have also added electron microscopy (Fig. 5), which is the first time description of ultrastructural changes during wing nerve injury.

In Figure 2A the authors show that mutated Atg8a^{G116*} which fails to undergo any lipidation event also leads to debris accumulation. It would be interesting to show with imaging whether at the site of injury Atg8a^{G116*} flies exhibit a diffuse cytosolic expression.

Thank you for this comment. We have already reported previously that the Atg8a^{G116} mutant is unable to form puncta in fat bodies and this truncated Atg8a protein shows diffuse cytosolic localization (Jipa Autophagy, 2021 Sep;17(9):2565-2575). Now we show that after TBI, endogenous Atg8a dots are present in control brains, but it remains diffuse cytosolic in Atg8a^{G116*} homozygotes (Fig. S3c).*

In Figure 2A and 2D the authors claim that the ATG1 complex is dispensable for axon debris removal although it is required for autophagic process. This statement is not supported by the findings provided. Firstly, the authors should show whether Atg101 and Atg17 are present in both injured and uninjured wing L1 veins. Next, it will be interesting to check whether loss of Atg101 or Atg17 disfavors the presence of Atg8-positive puncta at the site of injury

Thank you for this note. We have not initiated experiments on the presence of Atg101 or Atg17 in injured and uninjured wing L1 veins as these genes (just like all other non-redundant genes encoding core autophagy proteins) are universally expressed in all tissues including glia. The most recent high-throughput Drosophila study (Science. 2022 Mar 4;375(6584):eabk2432.) confirms this, showing cell type-specific data that even includes wing glia (reported in the paper). Here is an overview for the expression of these two genes from Flybase:

Atg17/FIP200:

Atg101:

We investigated the effect of the loss of *Atg101* on the number of *Atg8a*⁺ puncta in injured wing nerve glia by expressing *UAS-mCherry-Atg8a* specifically in glia. Please note that the site of injury cannot be imaged in the wing nerve due to strong melanization at the cut site. We thus always image regions further away from the site of injury where axon debris phagocytosis takes place. There are fewer, smaller, and fainter *mCherry-Atg8a* structures in the uninjured *Atg101* mutant wing nerve compared to control (Fig. 2d, e), in line with the already published autophagy defect of this mutant (Guo *J Biol Chem*, 2019 Apr 5;294(14):5666-5676). Injury clearly increased the size and brightness of *mCherry-Atg8a* structures in *Atg101* mutant wing glia (please see **Figure 2 for Reviewers**). As *Atg101* deficiency blocks canonical autophagy, these changes in the *Atg101* mutant indicate the upregulation of LAP.

Figure 2 for Reviewers. Both the size and brightness of glial mCherry-Atg8a puncta increase in injured Atg101 mutant wing nerves.

Figure 2D and 3E: The authors show that blockade of autophagy components leads to reduction of *Atg8a* positive puncta and an increase in Ref(2)P puncta in uninjured neurons. It would be interesting to show what happens on *Atg8a* and Ref(2)P puncta in control flies upon injury. Do they respond? As autophagy is not induced and LAP takes over, *Atg8a* should increase and Ref(2)P should stay static. Is this the case?

Thank you for this note. We have never investigated changes in Atg8a⁺ or Ref(2)P⁺ structures specifically in neurons, because our story focuses on LAP in glia. Please note that Fig. 3E shows Ref(2)P accumulation in glia (not in neurons) upon glial-specific knockdown of Atg genes. We thus assume the Reviewer meant to suggest doing these experiments in glia. We expressed either

Ref(2)P-GFP or mCherry-Atg8a in glia and injured the wing nerve. We saw an increase in mCherry-Atg8a+ puncta numbers post-injury (Fig. S3f). In contrast, Ref(2)P-GFP+ puncta numbers did not change (Fig. S3g). So yes, this is the case.

In Figure 2H the authors state that LAP and canonical autophagy act through the same pathway for debris engulfment. To make a definitive statement that couples the LAP with the canonical autophagy, more evidence has to be provided. In my view, a time course analysis of debris elimination at days 2, 5 and 10 dpi should be included upon single and double inhibition of *drpr* and *Atg8*. Moreover, double mutants of *drpr* with *Atg101* and *Atg17*, and subsequent examination of remaining axon debris will give additional information regarding the mode of interaction between LAP and autophagy.

Thank you for this note. We have not stated in the previous manuscript version that LAP and canonical autophagy act through the same pathway for debris engulfment. We wrote that „...Drpr and Atg8a likely participate in the same pathway underlying debris removal”, referring to Atg8a as being part of the LAP machinery in this context. We thus did not see a reason to „make a definitive statement that couples the LAP with the canonical autophagy” as put forth by the reviewer. We reasoned that analysis of any interaction in debris clearance at 2 dpi is futile as Atg loss does not have a significant effect on debris abundance at this early time point (please see Fig. 1) and analysis of Atg8a and drpr/+ genetic interaction at 10 dpi (Figure 3 for Reviewers) was similar to the 5 dpi data (now Fig. S4a). We then investigated the interaction of drpr/+ with null mutants for Atg101, Atg5 and the Atg8a^{G116} truncation allele at 5 dpi (please note that all Drpr interactors are now shown in Fig. S4). At 5 dpi, only Atg8a^{d4}; drpr/+ wing nerves showed significantly less debris compared to the Atg mutant alone. As expected, Atg101; drpr/+ wing nerves accumulated axon debris, unlike Atg101 alone (Fig. S4e). These also support that Atg5 and Atg8a function in LAP with Draper, while Atg101 does not. Please see also the response to the next comment for interpretation of the interaction of LAP genes with drpr/+.*

Figure 3 for Reviewers. Left: Single optical slices of degenerating wing nerves of the indicated genotypes on OK371> UAS-mCD8::GFP/+ background at 10 dpi. Right: Quantification of axon debris abundance in single-slice images of the indicated genotypes at 10 dpi.

Figure 2H,I: the results show that the combination of *drpr^{Δ5}* and *Atg8a^{Δ4}* actually reverse the effects of each single treatment (Although the quantification analysis shows a statistical difference between . This could indicate that they act redundantly in the clearance of axonal

debris. Is LAP activated when Drpr is mutated and vice versa? What is the sequence of events? Do they happen simultaneously?

Thank you for this comment. We have addressed this issue by carrying out new experiments using Atg5^{5cc5}, Atg8a^{G116} and Rubicon^{fs} mutants that all abrogate LAP, and tested their interaction with drpr^{Δ5/+}. There was no statistically significant decrease in axon fragment numbers in any of these cases compared with single Atg or Rubicon mutants (Fig. S4b-d), unlike in drpr^{Δ5/+};Atg8a^{Δ4} animals (Fig. S4a). Thus, we think that Atg8a likely has a lipidation- and autophagy- and LAP-independent role in this context as well, just like in other contexts shown previously by others and us (Xu Cell Death and Differ, 2015 Nov;22(11):1792-802, Jipa Autophagy, 17:9, 2565-2575). It has been clearly established that Draper is absolutely required for glial engulfment of all axon fragments (there is no engulfment in homozygous Draper null mutants, that is why we used heterozygotes for these experiments), and we show here that axon fragments are taken up into glial phagosomes, as both early (glial Rab5+) and late (glial Rab7+ glial Rubicon+) phagosomes contain engulfed axon debris. Since the Rab7 vesicles are also positive for glial-expressed Rubicon, these data clearly delineate the sequence of events.*

In the experimental design in Fig.4 authors should have used the Atg16MI mutants and not Atg16 RNAi that would impact both autophagy and LAP.

This experiment was specifically designed to manipulate LAP only in glia hence the use of Atg16 RNAi. We could not obtain recombinants expressing nSyb-lexA.DBD::QF.AD, lexAop2-CD4::tdTomato on an Atg16MI mutant background that is needed for this experiment. Instead, please see our new ultrastructural analysis in Fig. 5 that verifies the generation of phagosomes during this process.

In the same Figure, authors don't show whether the depicted debris is indeed engulfed by phagosomes.

Fig. 4 illustrates that glial lysosomes (here, GFP-Lamp1 is only expressed in glia!) contain axon debris that were generated outside of glia (here, mtdTomato is only expressed in neurons!). It has been firmly established that engulfment of axon fragments requires the phagocytic receptor Draper via phagocytosis (MacDonald Neuron, 2006 Jun 15;50(6):869-81, Logan Nat Neurosci, 2012 Mar 18;15(5):722-30.). Nevertheless, we provide further evidence in the revised manuscript, showing that axon fragments are indeed found within phagosomes derived from the glial membrane (Fig. S7e), and are present in glial Rab5+ early phagosomes as well as glial Rab7+ late phagosomes in both control and Atg16 depleted glia after injury (Fig. S8).

There is no direct evidence demonstrating that phagosomes contain axonal debris. Authors should use electron microscopy or specialized fluorescence microscopy methods to convincingly show this. Electron microscopy would be optimal as, through it, the authors could additionally show that debris are contained by a single membrane formation-as expected for the phagosome- and not in double membrane formations (that would be autophagosomes), further strengthening their initial observations that autophagy is dispensable.

Thank you for this comment. We have added ultrastructural analysis that demonstrate these in Figure 5. Please see our detailed response to the only comment of Reviewer 2 for details.

The claim that 2 days post injury a different type of Rubicon-Rab7 double positive vesicle appears only taking into account the size is unsubstantiated. The same stands for the claim that this differentiated type of vesicle contains less Rubicon due to limited Rubicon association. This could just be due to decreased protein expression in the injured compared to the uninjured. To claim that a different type of vesicle is formed authors should have isolated vesicles from the two time points and perform proteomic analysis to check their protein constituents or/ additionally they could have manually checked the presence or absence of endosomal markers in these (larger) structures.

Thank you for this comment. We agree with the reviewer about the uncertainty regarding the differentiated type of vesicles and thus we have toned down the interpretation of these experiments. Given the size increase of these vesicles in glia and their colocalization with axon fragments after wing nerve injury, we now suggest that incoming, phagocytosed axon debris contributes to the enlargement of these Rab7+ Rubicon+ vesicles after wing nerve injury.

Fig. 6A: The dotted expression pattern in this figure is not sufficient to support that Rubicon is localized to vesicles. What kind of vesicles? No vesicular marker is used to support this statement. Colocalization of glial Rubicon::mRFP1 vesicles and GFP-labelled axon debris does not seem prominently affected, when comparing wing nerve injury and random Rubicon puncta colocalization on GFP-labelled axons without injury. Puncta colocalization quantification is missing. Zoomed areas could be accompanied by single channels images in order to closely assess and evaluate the morphology of colocalized puncta.

Thank you for this comment. The former Fig 6a and 6b, (now we swapped these, see Fig. 7a and 7c) complement each other. Rab7 is a firmly established marker of late endosomes and late phagosomes in Drosophila (Sriram J Cell Biol, 2003 May 12;161(3):593-607, Akbar 2011, J Cell Biol 2011 Feb 7;192(3):383-90.). Fig. 7a demonstrates that Rubicon nearly completely colocalizes with late endosomes/phagosomes decorated with Rab7. Thus, Rubicon dots in Fig. 7c mostly correspond to Rab7+ late endosomes/late phagosomes. We have added puncta colocalization quantification for Rubicon::mRFP vesicles and GFP-labelled axon debris (Fig. 7d) based on criteria that we describe in Methods. As suggested, single channel panels are now shown, depicting zoomed areas of images in Fig. 7a, c.

Another issue relevant to Figure 6A and 6B pertains to the effect of injury in the total protein levels of Rubicon in the glia. While in Figure 6A Rubicon is increased in injured versus uninjured the opposite is shown in Figure 6B.

Thank you for pointing out this confusing image selection. We acknowledge the previous discrepancy between the two panels: the original Fig. 6a 2dpi was selected to illustrate colocalization and it was not representative for Rubicon level. We have thus included a new image in now Fig. 7a, 2 dpi that better represents the changes quantified in Fig. 7d.

The authors should show mechanistically whether Rubicon blocks endocytosis and if yes, then how are the rest of the results (regarding the effect of Rubicon on debris elimination) explained? What is the effect of Rubicon on LAP in glial cells?

Thank you for this note. We are afraid that there has been a misunderstanding with respect to the endocytic versus phagocytic (these are different processes!) nature of axon debris degradation and the role of Rubicon in that. The effect of Rubicon on LAP in glial cells appears to be the same as in any other cell that undergoes LAP: it facilitates phagosome conversion into a LAPosome that can fuse more efficiently with the lysosome (JCS 2019 Feb 20;132(5):jcs222984). LAP is not endocytosis as it has been described during phagocytosis in macrophages (Nature 2007 Dec 20;450(7173):1253-7., PNAS 2011 Oct 18;108(42):17396-401., Nat Cell Biol 2015 Jul;17(7):893-906), in the retinal pigment epithelium (Cell 2013 Jul 18;154(2):365-76), in Sertoli cells (Nat Commun 2017 Sep 19;8(1):598), in the ovarian granulosa cells (Hum Reprod 2020 Jun 1;35(6):1346-1362) and in C. elegans embryos (JCS 2016 Oct 15;129(20):3721-3731, Cell Rep 2018 May 15;23(7):2070-2082). It has been firmly established that Rubicon promotes phagocytic cargo degradation in lysosomes through LAP (Autophagy 2017;13(12):2072-2085, Nat Commun 2022 Oct 24;13(1):6233) at least in part by activating Vps34 on LAPosomes (Nat Cell Biol 2015 Jul;17(7):893-906). In contrast, Rubicon inhibits endosome processing and autophagosome maturation (MBoC 2010 Dec;21(23):4162-72., Nat Cell Biol 2009 Apr;11(4):468-76., Nat Cell Biol 2009 Apr;11(4):385-96., PNAS 2010 Nov 9;107(45):19338-43. Axon debris constitutes a phagocytic, and not an endocytic cargo as it is detected by the Draper phagocytic receptor and is engulfed into phagosomes, a mechanism that has been thoroughly studied (Nat Neurosci 2012 Jun 17;15(7):979-87., Neuron 2006 Jun 15;50(6):869-81). We did not investigate the effect of Rubicon on endocytosis and autophagy because these have already been demonstrated in at least four publications. The main finding of our work is a novel in vivo role of LAP: glial breakdown of degenerating axons. We would like to point out that our story is not focusing on Rubicon: Rubicon has a firmly established role in LAP, so we use this gene/protein (along with the other genes/proteins involved in LAP) as tools to show that LAP in glia is required for axon debris clearance in a complete animal, and we now show the physiological importance of LAP via analysis of a LAP-specific mutant (the Atg16 WD40 truncation) during brain injury (Fig. 9).

Minor

Most experiments performed do not include the control uninjured axons, it will be interesting and more informative to be included at least in the main experiments. The authors should include both positive and negative controls at least for the most essential experiments performed.

*We have added uninjured nerve images for all figures dealing with axon debris clearance. Please see Figs. S1b, S2, S4, S5a-g, S7a, S9f. The positive control for debris clearance was/is *drprΔ5/+* heterozygotes (now shown in Fig. S4). Baseline conditions or knockdown/knockout of an irrelevant gene is used as negative control.*

Figure 1B is not very informative. It should be combined with a bright field image so that the wing morphology becomes more recognizable.

Thank you for this note. We have added a combined DIC-fluorescent image to illustrate where the wing nerve is situated in this part of the wing in Fig. 1a.

In Figure 3A, contrary to what is mentioned in the text, *atg1* inhibition seems to cause debris removal defects.

Thank you for this note. In Fig. 3A (currently Fig. 3e), slight debris accumulation was occasionally seen in Atg1 RNAi based on quantification in Fig. 3B (currently Fig. 3f). Since the image was indeed not representative of the statistics, we have replaced this image with one that better matches the statistics.

In Figure 6A co-localization is not obvious. Author should perform statistical analysis

Thank you for this note. We have added quantification of GFP+ debris and Rubicon-mRFP colocalization at both time points (now Fig. 7d).

Figure 2E: It is not clear why the authors choose to use two different statistical tests for the comparison of Atg101 and Atg17/FIP200 mutants as opposed to Atg5 mutants with the control .

In pairwise statistical tests based on null hypothesis testing, defining the distribution of the data is required to decide which test can be applied. The t-test assumes normally distributed (Gaussian) datasets. If this assumption is not true for all datasets, a non-parametric test needs to be applied such as the Mann-Whitney test in the Atg5 mutant comparison, as described also in Nature Portfolio journals policy on statistics.

Figure 3a: Axonal debris are still visible in atg1 RNAi flies, with an incidence higher than in control flies. This phenomenon is not recapitulated in the quantification panel 3b. Maybe a different image should be used.

Thank you, this has already been answered in minor point 3 above.

Figure 7: The image of axonal debris in this strain is completely different than that in the mCD8::GFP strains. There are no obvious puncta or uninjured neurons. How do the authors explain this difference?

Thank you for mentioning this difference. We stated that the genotype used here is nSyb-LexA driven myr::GFP (nSyb> lexAop-myr::GFP). This driver labels all (about 200) neurons in the L1 vein as opposed to the OK371-Gal4 driver used in most figures (which labels approximately 40 neurons). The pan-neuronal driver thus labels all axons resulting in a much denser, packed appearance of axon debris until fragment breakdown kicks in. This is especially obvious at 2 dpi, an early time point shown here. Uninjured axons (which are also more numerous with the pan-neuronal driver) are partially masked by the dense debris. We used this labeling in now Fig. 8a to better capture the colocalization of mCherry-Atg8a with axon debris.

Reference 85 is missing.

Thank you pointing out this mistake. We corrected this error.

The introduction and discussion are rather long and could be more focused.

Thank you, we have improved the coherence and shortened these parts.

Reviewer #4 (Remarks to the Author):

In this manuscript, Szabó et al claim that glial LC3-associated phagocytosis (LAP) promotes clearance of degenerating axons. The *Drosophila* LC3 homologue is Atg8. They claim that LAP depends on Atg8 EP lipid conjugation, the Atg8 conjugation system formed of Atg5, Atg12 and Atg16, and on the Vps34 lipid kinase subunits UVRAG and Rubicon, but not Atg14 or the Atg1 kinase complex members Atg101 or Atg17. They show that knock-down in glia of UVRAG, rubicon or Vps34 all prevent debris clearance. They claim that Atg8 and Drpr work in the same pathway; that Atg8 and Rubicon are recruited to Rab7+ phagosomes; that Rubicon normally inhibits endocytosis, which is similar to phagocytosis, in un-injured samples; that knock-down of Rubicon in glia prevents debris clearance; that a decrease in Rubicon levels accompanies the maturation of LAPosomes; whereas over-expression of Rubicon in glia accelerates axon debris clearance.

The following points need resolving and suggestions are made for improvement:

1. Evidence unclear on the involvement of Atg8 in multiple aspects of study:

If the wing is cut and thrown away, as in Figure 1, how is it possible that ‘some axons are spared by the injury’ as stated in the methods section? Is it that some axons do not undergo Wallerian degeneration, and is this affected differentially by the different genotypes? The quantifications described in the methods say that a free-hand ROI excluding the axons is chosen to quantify axonal debris. However, remaining axons differ across genotypes, eg there seem to be more remaining axons in both Atg8 LOF and GOF in Figure 1. For example, remaining axons in Fig.1B vs control could suggest that Atg8 is required for Wallerian degeneration in axons. Quantifying the remaining axons could reveal whether Atg8 is required for Wallerian degeneration or not.

We thank the reviewer for pointing this out. While the wing nerve injury model is well-established in the field, it is important to make it clearer in the Introduction and Methods which axons do not degenerate. Now we explain this in the text, revised Fig. 1a and added a detailed drawing, depicting that there are axons that are not cut during wing transection, so these will not degenerate (Fig. S1a). This is because the wing is cut in the middle by using fine scissors, which is of course not happening with nanometer precision hence there is random variation in the number of uncut axons: the axons of neurons located closer to the body of the relative to the actual wing cut site are not severed. This leaves us with a small, somewhat variable number of uninjured axons that serve as landmarks for imaging the wing nerve (please see also the ultrastructure of wing nerve cross-sections in Fig. 5). There is no statistically significant difference in the number of remaining uninjured axons in control vs Atg8a⁴⁴ mutants (Fig. S1c).

A previous paper found that autophagy in neurons has a minor role in the Wallerian degeneration program. Upon neuron-specific knockdown of Atg12 or Atg17 (both genes are required for canonical autophagy), wing nerve axon fragmentation was slightly delayed after cutting the wing, which was only seen at 1 day post-injury (Sci Adv 2019 Feb 13;5(2):eaav4971) and this delay was gone by 3 days post injury. In comparison, axons lacking dSarm or axundead (core effectors of the Wallerian degeneration machinery) do not degenerate for weeks after injury (Neukomm et al., Neuron, 2017 Jul 5;95(1):78-91.e5). Thus, it is LAP in glia rather than axon-intrinsic autophagy that affects debris clearance in Atg8a⁴⁴ mutants, which we have now further

confirmed through the analysis of adult glia-specific knockdown of Atg8a using two independent RNAi lines (Fig. 3i,j).

Based on Figure 2I the authors conclude that Atg8 and Drpr function in the same pathway, because the double LOF *atg8 drpr-* gives a not-statistically significant difference vs Atg8 LOF. However: 1, the double is statistically significantly different from *drpr-/+* alone; 2, the sample size differs in Atg8D4 (n=16) vs the double (n=12); 3, the median in the double is virtually the same as for the control and differs greatly from the medians for both *drpr/-* and Atg8D4 mutants. Thus, the graph evidence indicates that the sample size for the double must be increased to at least n=16 as it could become statistically significantly different from Atg8D4 alone. This is important, as it could indicate that in the double *drpr-/+* Atg8D4 there is a rescue phenotype. Together with the above, it could indicate that Atg8 is required in axons for Wallerian degeneration.

Thank you for this comment. We have carried out similar experiments with Atg5^{5cc5}, Atg8a^{G116} and Rubicon^{fs1} mutants regarding their interaction with drpr^{45/+} heterozygotes. There was no statistically significant change in any of these combinations compared to single Atg5^{5cc5}, Atg8a^{G116*} and Rubicon^{fs} mutants (Fig. S4b-d). We have also increased the sample size of drpr^{45/+}; Atg8a⁴⁴ to n=20. Here, we do observe a statistically significant decrease compared to the Atg8a⁴⁴ mutant (Fig. S4a). However, this effect is unique because it is not seen in case of heterozygous drpr⁴⁵ combinations with Atg5^{5cc5} or Atg8a^{G116*} or Rubicon^{fs}. These experiments thus suggest a lipid conjugation-, autophagy- and LAP-independent role of Atg8a specifically. Please note that such non-canonical roles for Atg8a have already been reported in other contexts by others and us (Xu Cell Death and Differ, 2015 Nov;22(11):1792-802, Jipa Autophagy, 17:9, 2565-2575). Drpr heterozygosity does increase fragment accumulation in autophagy-defective Atg101 null mutants (which phenocopy control injured wings on a drpr+ background). These also suggest that Atg5, Atg8a and Rubicon function in LAP with Draper, while Atg101 does not. We have moved all these genetic interaction tests into the supplement (please see Fig. S4).*

The authors then test the involvement of the conjugation system Atg5 and Atg16 and of Atg1, Atg13 and Atg 17 by knocking them down in glia, and they conclude that the conjugation system is required but not Atg1 complex. However, they did not test Atg8. It is essential that they test *repoGAL4>UAS-Atg8-RNAi*.

We thank the reviewer for this suggestion. We have tested two independent RNAi lines against Atg8a using the drug-inducible Geneswitch (GS) system to knock down Atg8a in adult glia, which caused a clear and statistically significant axon debris accumulation after injury in case of both RNAi lines (Fig. 3i, j).

2. Evidence must be provided that RNAis and glial-targeted CRISPR/Cas9 are being knocked-down, with qRT-PCR. Instead, they use an autophagy assay as evidence of RNAi knock-down in experiments which address the question of whether autophagy is involved or not. This is indirect, circular and not appropriate, and the evidence of RNAi knock-down must be direct and specific to the targeted gene.

Thank you for this comment. We now include qRT-PCR measurements for RNAi transgenes. Because repo-Gal4 is only active in glia which constitute only 10% of the nervous tissue, we used

the ubiquitous tubulin-Gal4 for organism-wide knockdown, isolated RNA from whole animals and performed qRT-PCRs on these samples. We found statistically significant knockdown for all RNAi constructs tested (Fig. S6). Rubicon mRNA did not change in response to tubulin-Gal4 mediated CRISPR mutagenesis, which is not surprising because CRISPR induces small indels that do not necessarily cause reduction in transcript levels. Atg1 RNAi resulted in early lethality with tubulin-Gal4, but the efficiency of the same Atg1 RNAi construct was already shown by qRT-PCR in previous publications that we now cite in Methods: a reduction to 8% with a maternal tubulin-Gal4 driver (Dev Cell, 2014 Oct 13;31(1):114-27) and a reduction to about 20% with a midgut driver, NP1-GAL4 (Cell Death and Differ, 2012 Aug;19(8):1299-307).

3. The function of rubicon is unclear: if Rubicon normally inhibits endocytosis, and Rubicon signal intensity and puncta decrease after injury reflecting the maturation of LAPosomes, then why does rubicon RNAi prevent cell debris clearance, and over-expression increases it?

Thank you for this comment. We discuss this in detail in our response to the last major comment of Reviewer 3 (please see also our answer page 12). Since the size of Rubicon+ Rab7+ vesicles increases about 3-fold in glia after injury (which probably “dilutes” the Rubicon signal), we think that incoming phagocytosed cargo (axon fragments)/cargo receptor (Draper) activate Rubicon to promote LAPosome maturation and breakdown. We hypothesize that Rubicon’s inhibitory (endocytosis, autophagy) vs activating (LAP) function depends on the type of cargo/cargo receptor arriving to Rab7 vesicles. We now include these considerations in the revised Discussion.

4. Genetic background contribution to phenotypes must be ruled out by using trans-heterozygous combinations of 2 different alleles: Fig.1,2,3,5.

Thank you for this comment. In Fig 1, we added an Atg5^{5cc5}/Atg5^{GT14} combination as well as genetic rescue of Atg5^{5cc5} by a genomic Atg5 rescue transgene, both of which rules out background contribution. In Fig 2, we added the Atg16^{Mi}/Atg16^{A67} combination. We added the analysis of two independent Rubicon frameshift alleles in trans in previous Fig 5/now Fig 6. We show the rescue of the Atg8a^{A4} phenotype by a genomic rescue transgene at 2 different time points (Fig. 1b, c), which again rules out the contribution of genetic background. Furthermore, we included two independent RNAi lines knocking down Atg8a specifically in glia that phenocopy the Atg8a mutant results (Fig. 3i, j).

5. Statistical analysis for figures that contain graphs with more than 2 genotypes (e.g., Fig.1I, Fig.2B etc.): it would appear the p values on the figures correspond to the multiple comparisons correction test results? This information must be provided in the figure legends. They should include in the figure legend the p value for the Kruskal Wallis ANOVA and/or One Way ANOVA, the multiple comparisons corrections tests used and then indicate what test the p values on the graphs refer to.

Thank you for this comment. We now provide full statistical information in figure legends regarding the statistical test used, the overall p-value if there is any statistical difference in the test, the multiple comparison correction method and individual p values based on the multiple comparison correction test.

6. Writing could improve. From the background information in the introduction, the novelty of this study is not always clear. Too much background information is also provided within results, in places.

Thank you for this note. We improved these aspects of the manuscript, although we had to add some further explanations in the Results section based on the Reviewer comments.

7. An illustration figure of the pathways under study would really help.

Thank you for this note. We have included a schematic of the pathway in Figure 10.

Applies to most figures:

1. Non-injured controls must be provided in figures, as evidence that injury took place.

Thank you for this note. We have added non-injured control images of contralateral (non-cut) wings in Supplemental Figures: please see Figs. S1b, S2, S4, S5a-g, S7a, S9f.

2. Would be best to avoid writing the genotype over image data as they obscure the data in multiple cases, specially with long genotypes.

Thank you for this note, we have moved all text so that they do not overlap with confocal images.

3. Images are given as Z-projections, and quantifications are mostly given of single sections. Thus, images are not representative of the data given in the graphs. Instead, quantifications must reflect the image data provided, and vice-versa. If the quantifications are on single sections, then the images ought to be too.

Thank you for this note. We have replaced all microscopy images with single sections.

4. Figures must contain complete genotypes, currently some (eg Figure 2D,E) do not.

Thank you for this note. This has been corrected.

5. Figures claiming co-localisation must include split channels of their higher magnification views, otherwise evidence is not visible: Figure 4, 6, 7.

Thank you for this note. Split channels have been added to the relevant panels.

REVIEWERS' COMMENTS

Reviewer #1 (Remarks to the Author):

The authors have full addressed my concerns. In my reading of the other three reviews, I further think the concerns raised by all reviewers have been adequately addressed. I support publication of this paper in its current form.

Reviewer #2 (Remarks to the Author):

The authors have satisfactorily addressed my concerns

Reviewer #3 (Remarks to the Author):

The authors have adequately responded to the comments of the Reviewers and the manuscript has now been improved substantially.

Reviewer #4 (Remarks to the Author):

The authors have made a tremendous effort to improve the manuscript, provided abundant new data, changed figures and improved the text. The authors have addressed all the experimental concerns I had raised. I am satisfied with the revision and I would like to congratulate the authors on their much improved paper.

Response to reviewer comments

Reviewer #1 (Remarks to the Author):

The authors have fully addressed my concerns. In my reading of the other three reviews, I further think the concerns raised by all reviewers have been adequately addressed. I support publication of this paper in its current form.

Response: Thank you for supporting the publication of our manuscript.

Reviewer #2 (Remarks to the Author):

The authors have satisfactorily addressed my concerns

Response: Thank you for the positive evaluation of our manuscript.

Reviewer #3 (Remarks to the Author):

The authors have adequately responded to the comments of the Reviewers and the manuscript has now been improved substantially.

Response: Thank you for the positive evaluation of our manuscript.

Reviewer #4 (Remarks to the Author):

The authors have made a tremendous effort to improve the manuscript, provided abundant new data, changed figures and improved the text. The authors have addressed all the experimental concerns I had raised. I am satisfied with the revision and I would like to congratulate the authors on their much improved paper.

Response: Thank you for the positive evaluation of our manuscript.